# A small molecule that mitigates bacterial infection disrupts Gram-negative cell membranes and is inhibited by cholesterol and neutral lipids

Jamie L. Dombach[1]*, Joaquin L. J. Quintana[1], Toni A. Nagy[1], Chun Wan[1], Amy L. Crooks[1¤], Haijia Yu[1], Chih-Chia Su[2], Edward W. Yu[2], Jingshi Shen[1], Corrella S. Detweiler[1]*

1 Department of Molecular, Cellular, and Developmental Biology, University of Colorado, Boulder, Colorado, United States of America, 2 Department of Pharmacology, Case Western Reserve University, Cleveland, Ohio, United States of America

¤ Current address: Array BioPharma/Pfizer, Boulder, Colorado, United States of America
* jamie.dombach@colorado.edu (JLD); detweile@colorado.edu (CSD)

**Data Availability Statement:** All relevant data are within the manuscript.

## Abstract

Infections caused by Gram-negative bacteria are difficult to fight because these pathogens exclude or expel many clinical antibiotics and host defense molecules. However, mammals have evolved a substantial immune arsenal that weakens pathogen defenses, suggesting the feasibility of developing therapies that work in concert with innate immunity to kill Gram-negative bacteria. Using chemical genetics, we recently identified a small molecule, JD1, that kills *Salmonella enterica* serovar Typhimurium (*S.* Typhimurium) residing within macrophages. JD1 is not antibacterial in standard microbiological media, but rapidly inhibits growth and curtails bacterial survival under broth conditions that compromise the outer membrane or reduce efflux pump activity. Using a combination of cellular indicators and super resolution microscopy, we found that JD1 damaged bacterial cytoplasmic membranes by increasing fluidity, disrupting barrier function, and causing the formation of membrane distortions. We quantified macrophage cell membrane integrity and mitochondrial membrane potential and found that disruption of eukaryotic cell membranes required approximately 30-fold more JD1 than was needed to kill bacteria in macrophages. Moreover, JD1 preferentially damaged liposomes with compositions similar to *E. coli* inner membranes versus mammalian cell membranes. Cholesterol, a component of mammalian cell membranes, was protective in the presence of neutral lipids. In mice, intraperitoneal administration of JD1 reduced tissue colonization by *S.* Typhimurium. These observations indicate that during infection, JD1 gains access to and disrupts the cytoplasmic membrane of Gram-negative bacteria, and that neutral lipids and cholesterol protect mammalian membranes from JD1-mediated damage. Thus, it may be possible to develop therapeutics that exploit host innate immunity to gain access to Gram-negative bacteria and then preferentially damage the bacterial cell membrane over host membranes.

**Funding:** The work was supported by National Institutes of Health grants AI126453 (CSD), AI121365 (CSD), GM126960 (JS), and AI145069 (EWY) https://www.nih.gov. The funders did not play any role in study design, data collection and analysis, the decision to publish, or preparation of the manuscript.

**Competing interests:** The authors have declared that no competing interests exist.

## Author summary

Bacteria are increasingly becoming resistant to the antibiotics that are currently available. It has even been predicted that in the next thirty years there will be more deaths from antibiotic resistant infections than from cancer. We therefore need new antibiotics. To decrease the likelihood that bacteria will rapidly develop resistance to new antibacterials, researchers are seeking novel compounds that work differently than existing antibiotics. To find such a compound, we looked for chemicals that reduce the number of infectious bacteria within mammalian cells. We focused our efforts on Gram-negative bacteria because this class of pathogens is particularly difficult to treat with antibiotics. We present data showing that the compound JD1 disrupts bacterial cell membranes, a structure not targeted by current antibiotics for Gram-negative bacteria. JD1 also decreases bacterial colonization of infected mice. This is the first compound, to our knowledge, that preferentially targets the cell membranes of Gram-negative bacteria and reduces bacterial infection of animals.

## Introduction

Gram negative bacterial pathogens are equipped with numerous defenses that make them inherently difficult to treat. The presence of an outer membrane barrier prevents many antibiotics from accessing the cell. This barrier is maintained by lipopolysaccharides (LPS) in the external leaflet of the outer membrane and by efflux pumps that span both membranes and the periplasm [1]. LPS consists of three major components: the surface O-antigen polysaccharide, the saccharide core, and the membrane bound lipid A. Disruption of any component sensitizes the outer membrane and reduces its ability to exclude compounds [2]. Efflux pumps capture toxic molecules from the periplasm or the cytosol, including antibiotics and host antimicrobial peptides (AMPs), and export them across the outer membrane [3,4], contributing to antibiotic resistance and virulence [5]. Many clinical isolates of multidrug resistant bacteria have acquired mutations or genes that increase the number or activity of efflux pumps [6]. Thus, a major hurdle for antibiotic discovery in Gram-negative bacteria is the identification of compounds that can cross the outer membrane barrier and remain within the cell [7].

Another bottleneck for antibiotic discovery is the need to identify new drug targets; there is widespread resistance to antibiotics with established targets, such as ribosomes and DNA gyrase [8]. One possible underexploited target for Gram-negative bacteria is the cell membrane [9]. Bacterial cell membranes differ from mammalian membranes in their lipid composition and the absence of cholesterol [10]. Gram-negative bacterial membranes have an overall more negative charge due to the presence of phosphatidylglycerol (PG) and cardiolipin (CL; diphosphatidylglycerol) [11]. Mammalian membranes are composed mostly of neutral lipids, such as phosphatidylcholine (PC) and phosphatidylethanolamine (PE), but contain some negatively charged phosphatidylserine (PS) [10]. Cholesterol, at 10% of mammalian lipid composition, decreases membrane fluidity at physiological temperatures [12]. The differences between bacterial cell membranes and mammalian membranes suggest that chemicals that preferentially damage the former over the latter may have utility for combatting bacteria [10].

A third challenge of developing new potential antibiotics is that during infection, pathogens are exposed to complex microenvironments that are distinct from standard laboratory microbiological media [13,14]. For example, soluble host molecules, such as serum complement and proteases, damage the Gram-negative bacterial outer membrane [15]. Some Gram-negative

bacteria, including *Salmonella enterica* serovar Typhimurium (*S.* Typhimurium), reside and replicate within host cell vesicles called phagosomes. *S.* Typhimurium causes a natural, systemic, murine infection by multiplying within phagosomes of cells of the monocyte lineage [16]. Within phagosomes, *S.* Typhimurium is exposed to particularly harsh conditions, including various AMPs, proteases, lysozyme, low pH, and limited nutrients [17,18]. AMPs and magnesium limitation specifically compromise the LPS surface O-antigen polysaccharide in the outer membrane, increasing bacterial susceptibility to other host insults [15,19]. The complexity of the microenvironment that *S.* Typhimurium and other pathogens experience during infection is difficult to replicate in the laboratory but requires consideration for the purpose of discovering new chemicals with antimicrobial potential.

To identify chemicals that may have antibacterial activity during infection, we screened a library for compounds that prevent *S.* Typhimurium accumulation in a macrophage-like cell line (RAW 264.7). We developed the high-content screen, SAFIRE (Screen for Anti-infectives using Fluorescence microscopy of IntracellulaR Enterobacteriaceae), which monitors macrophage vitality via mitochondrial and nuclear staining and reports the accumulation of bacteria within macrophages on the basis of GFP expression. Macrophages were infected and two hours later treated with compound for 16 hours. Data were acquired using automated imaging. Compounds that reduced GFP expression in the SAFIRE assay were screened for their ability to reduce bacterial colony forming unit (CFU) recovery from macrophages. This approach identified from the 14,400 compounds in the Maybridge HitFinder v11 library 58 small molecules that enable the killing of *S.* Typhimurium within macrophages but not in in standard microbiological media. Three compounds from this screen inhibit bacterial efflux pumps, and another stimulates autophagy in macrophages [20–22].

Here we focus on a compound, JD1, that reduces *S.* Typhimurium growth and/or survival in macrophages by approximately 95%. In addition, JD1 inhibits bacterial growth under broth conditions that weaken the bacterial outer membrane or compromise efflux. We found that JD1 fluidizes and disrupts the cell membrane of bacteria, and damages liposomes that have a lipid composition similar to that of bacterial cell membranes. Liposomes with compositions that mimic mammalian membranes appear to be protected from the effects of JD1 by a combination of neutral lipids and cholesterol. We also show that JD1 has efficacy in a mouse model of *S.* Typhimurium infection.

## Results

### A small molecule prevents the survival of *S.* Typhimurium in macrophages

JD1 is a small aromatic molecule that contains a piperidinepropanol core with an adamantyl group, a favored component of drugs because it is generally stable, non-reactive, minimally toxic and increases solubility (Fig 1A) [23]. JD1 has not been described previously as having biological activity. In the SAFIRE assay, treatment of infected RAW 264.7 cells with JD1 reduced the accumulation of *S.* Typhimurium within macrophages by 95% at 1.2 μM, with a half maximum inhibitory concentration ($IC_{50}$) value of 0.12 μM (Fig 1B and 1C). Live imaging of infected macrophages treated with a range of JD1 doses revealed that as little as 0.06 μM significantly prevented, and 0.6 μM completely blocked accumulation (Fig 1G and S1–S6 Videos). To establish whether treatment with JD1 reduces bacterial load and not just GFP signal, we plated lysed macrophages for bacterial CFU, which declined with increasing dose (0.19 μM $IC_{50}$) (Fig 1D). Infection experiments with the human epithelial HeLa cell line similarly demonstrated JD1 antibacterial activity (1.77 μM $IC_{50}$) (Fig 1E and 1F). We also note that two different *S.* Typhimurium strains responded similarly to JD1 treatment in the context of infection

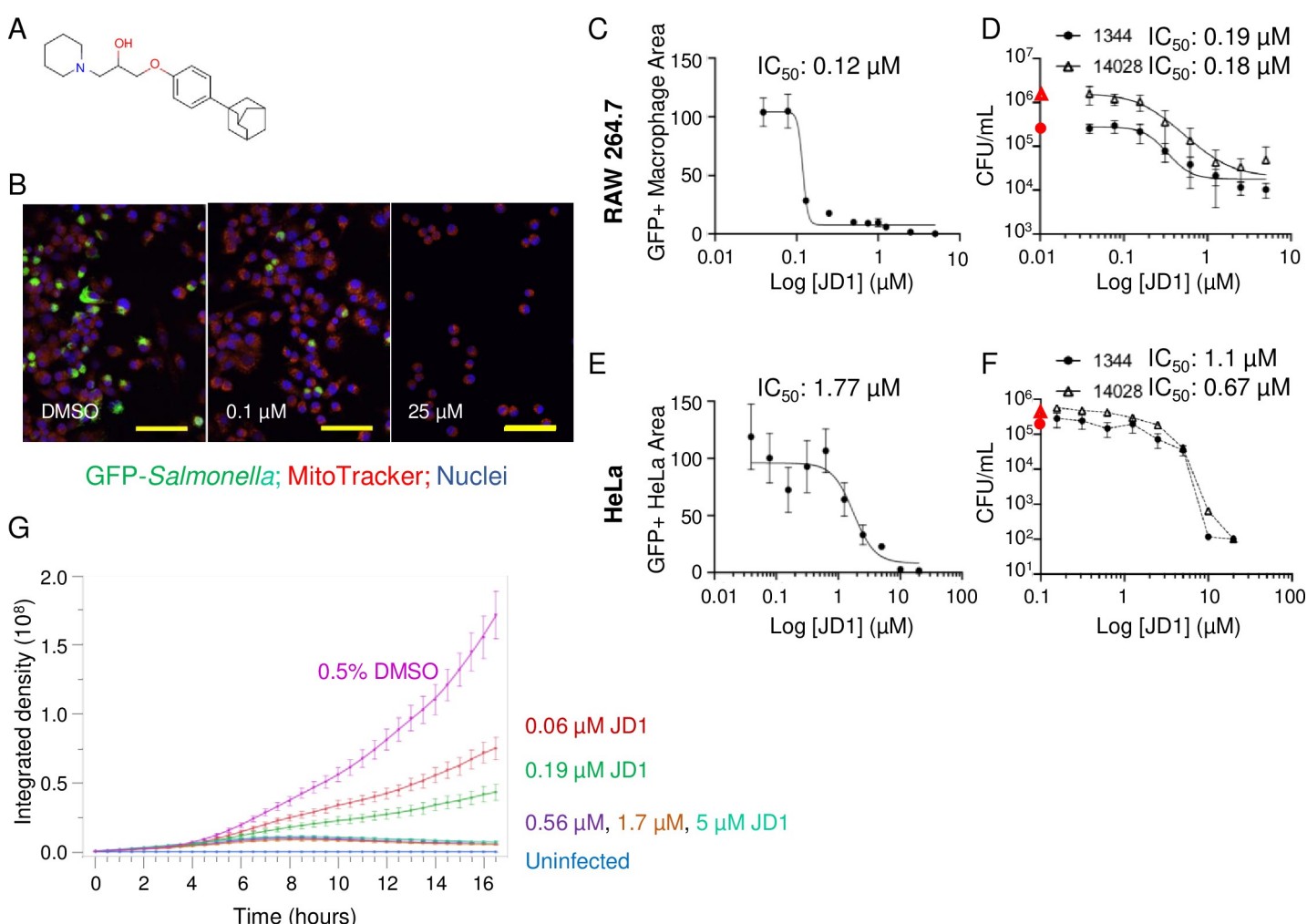

**Fig 1. Treatment of *S*. Typhimurium-infected macrophages and HeLa cells with JD1 prevents bacterial replication and/or survival.** A) Structure of JD1. B-D, G) RAW 264.7 macrophage-like cells were infected with *S*. Typhimurium harboring a chromosomal *sifB*::*gfp* reporter or E, F) HeLa cells were infected with *S*. Typhimurium harboring a chromosomal *rpsM*::*gfp* reporter. B-G) Cells were treated two hours after infection with vehicle (DMSO) or JD1 as indicated. After 18 hours of infection, cells were (B, C, E) fixed and imaged or (D, F) lysed and plated for enumeration of CFU. B) Representative micrograph of cells treated with DMSO (left), 0.1 μM JD1 (center), or 25 μM JD1 (right). Scale bars are 63 μm. C, E) GFP+ Macrophage/HeLa Area (as percent of DMSO) quantified from micrographs of cells treated with dilutions of JD1 from 5 μM for RAW 264.7 or 20 μM for HeLas. GFP+ Macrophage/HeLa Area is defined as the number of GFP-positive pixels per cell divided by the total number of pixels per cell, averaged across all cells in the field. Mean and SDs of technical duplicates from one of two biological replicates across 10 dilutions of JD1. The $IC_{50}$ value is indicated. D, F) CFU/mL of cells treated with dilutions of JD1 from 5 μM for RAW 264.7 or 20 μM for HeLas infected with *S*. Typhimurium strain SL1344 or 14028 as indicated. The red symbol on the Y-axis is the CFU value from DMSO-treated samples. Mean and SDs of biological duplicates each performed in triplicate with 9 dilutions of JD1. The $IC_{50}$ value is indicated. G) Live imaging of infected macrophages. Time 0 is 2 hours after infection, when compound or DMSO control were added. The integrated density is defined as signal obtained from maximum intensity projections of GFP+ Macrophage Area across six microscope fields. Data presented are the mean and SEM of three biological replicates each performed with technical triplicates. Uninfected cells show GFP baseline.

(Fig 1D–1F). The data thus indicate that the antimicrobial effects of JD1 are not restricted to macrophages, to murine cells, nor to a single bacterial strain.

To determine whether JD1 may synergize with gentamicin, the extracellular antibiotic standardly used in cell culture *S*. Typhimurium infection assays [24], we performed experiments in broth and in macrophages. A checkerboard growth assay in LB with 0.5 μg per mL of polymyxin B (see the next results section) did not reveal synergy between JD1 and gentamicin (S1A Fig). In cell culture infection assays, gentamicin was added 45 minutes after infection to prevent the growth of bacteria that were not engulfed by the cells. The gentamicin addition is necessary to prevent rampant extracellular overgrowth and cannot be replaced by washing.

The lowest concentration of gentamicin that prevents sufficient extracellular growth in SAFIRE assays is 40 μg/mL [21]. To establish whether gentamicin synergizes with JD1 during infection of macrophages, we added 40 μg/mL of gentamicin to RAW264.7 cells 45 minutes after infection and then either kept the concentration at 40 μg/mL or at two hours after infection reduced it to 10 μg/mL for the duration of the experiment. The lower concentration of gentamicin did not significantly alter the GFP+ macrophage area values in the presence of JD1 (S1B Fig). In the CFU assay, there were more colonies in 10 μg/mL gentamicin at all concentrations of JD1 (S1C and S1D Fig). Together the results suggest that gentamicin reduces the total CFU largely by decreasing number of extracellular bacteria, many of which are not counted by the SAFIRE method. These data do not support the notion that gentamicin potentiates JD1 activity in macrophages. We conclude that JD1 treatment of infected macrophages and HeLa cells prevents the replication and/or survival of *S.* Typhimurium.

## Conditions that compromise the bacterial outer membrane enable JD1 to inhibit growth and kill bacteria

Despite clear antibacterial activity in macrophages, JD1 did not reduce *S.* Typhimurium growth in LB medium at concentrations up to 150 μM, near the limit of JD1 solubility (Fig 2A and Table 1). To enable study of the mechanism(s) of JD1 antibacterial activity, we sought growth conditions that potentiate the compound. We therefore established whether media that mimic the macrophage phagosome, nutrient limitation and low magnesium [17,18], sensitize bacteria to JD1. Bacteria were grown overnight in nutrient poor M9 medium with 1 μM magnesium (M9-lowMg), which de-stabilizes the LPS polysaccharide layer [25]. Bacteria were diluted into the same medium. Treatment with JD1 inhibited *S.* Typhimurium growth at 77 μM, 1x MIC (defined as the concentration at which 95% of growth was inhibited). Similarly, the MIC of JD1 was 89 μM in a nutrient poor, acidic, low-phosphate, low-magnesium medium (LPM) specifically designed to resemble the phagosome environment [26–28]. It appears that nutrient limitation and limited cations modesty facilitate JD1 activity against *S.* Typhimurium.

We also determined whether an *E. coli* mutant strain frequently used to evaluate cell envelope stability could contribute to our understanding JD1 activity. The K12 *lptD4213* strain has a loss-of-function mutation in the gene encoding LptD/RlpB/Imp, which shuttles LPS to the outer leaflet of the outer membrane [29–31]. This strain therefore has a more permeable outer membrane [32–34] and is sensitive to antibiotics and detergents [29]. We found that the parent K12 strain was slightly inhibited for growth at 150 μM JD1 in LB. In contrast, the *lptD4213* mutant strain in LB was more sensitive to JD1, which had an MIC of 26 μM (Fig 2B and Table 1). Thus, sensitivity to JD1 may be increased by outer membrane permeability in the *lptD4213* mutant strain, a useful tool for understanding JD1 activity.

During infection of macrophages, bacterial outer membrane permeability is likely compromised by cationic antimicrobial peptides (cAMPs), which are ubiquitous in body fluids and are also present in phagosomes [17,18,35]. Polymyxin B (PMB) is a cAMP that at 0.5 μg/mL permeabilizes the *S.* Typhimurium outer, but not the inner membrane [25]. Polymyxin B non-apeptide (PMBN) is an attenuated derivative of PMB lacking the fatty acid tail. PMBN is less efficient than PMB at damaging the outer membrane [36], does not produce hydroxyl radicals [37] and is not, like PMB, reported to have other activities [38]. We grew bacteria overnight in LB and diluted them into LB with 0.5 μg/mL PMB or 10 μg/mL of PMBN prior to adding JD1. In LB with PMB, JD1 had an MIC of 14 μM, whereas in PMBN, the MIC of JD1 was nearly 10-fold higher (Fig 2A and Table 1). The addition of the iron chelator deferasirox did not rescue growth (S2A Fig), indicating that hydroxyl radical formation by PMB did not contribute to JD1-mediated growth inhibition [39]. Thus, PMB is a better potentiator of JD1 than PMBN,

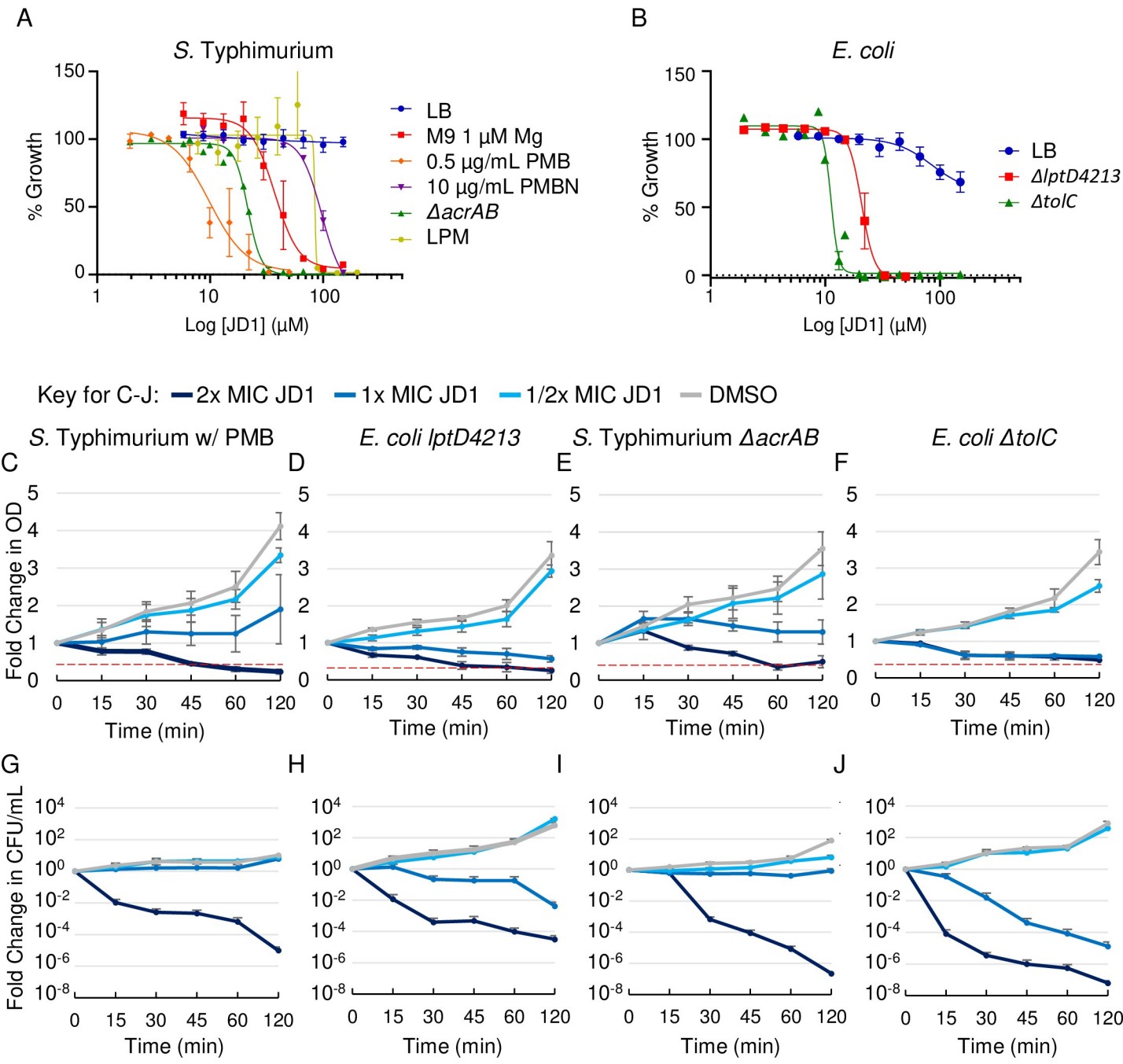

**Fig 2. JD1 is bacteriostatic and bactericidal under conditions that compromise the outer membrane barrier.** A, B) Dose response curves monitoring bacterial growth under the indicated conditions/strains normalized to DMSO for A) *S.* Typhimurium and B) *E. coli* K12. Data are normalized to growth in DMSO (100%). Mean and SEM of at least three independent biological replicates performed with technical triplicates. C-J) Log phase cultures of the indicated strains/conditions were treated at time 0 with either DMSO or the corresponding MIC$_{95}$ concentration of JD1 (Table 1). (C-F) Cultures were monitored for OD$_{600}$. The red dotted line denotes the limit of detection. (G-J) Cultures were also plated for enumeration of CFU. Mean and SEM of three biological replicates performed with technical triplicates. The medium used was LB unless otherwise indicated next to the strain name.

perhaps on the basis of outer membrane damage. We conclude that the antibacterial activity of JD1 against *S.* Typhimurium can be studied in LB with PMB.

Dissection of JD1 mechanism(s) of action in *S.* Typhimurium and *E. coli* requires knowledge of whether, and at which dosages, this compound kills bacteria. We therefore plated

**Table 1. Concentrations of JD1 that inhibit bacteria under different conditions.**

| Species (strain) | Genotype | Growth conditions (read-out) | IC$_{95}$ | |
|---|---|---|---|---|
| | | | μM | μg/mL |
| *S.* Typhimurium (SL1344) | WT | In macrophages (GFP+ Macrophage Area) | 1.2 | 0.49 |
| | WT | In macrophages (CFU enumeration) | 1.3 | 0.53 |
| | WT | In HeLas (GFP+ Macrophage Area) | 5.5 | 2.24 |
| | WT | In HeLas (CFU enumeration) | 7.1 | 2.89 |
| **Species (strain)** | **Genotype** | **Growth conditions (OD$_{600}$)** | **MIC$_{95}$**[1] | |
| | | | μM | μg/mL |
| *S.* Typhimurium: The strain background is SL1344 (WT) unless otherwise indicated | WT | LB | >150 | >61.0 |
| | 14028 | LB | >150 | >61.0 |
| | WT | LB, 0.5 μg/mL PMB | 14.1 | 5.7 |
| | 14028 | LB, 0.5 μg/mL PMB | 15.9 | 6.5 |
| | WT | LB, 10 μg/mL PMBN | 137.8 | 56.3 |
| | WT | M9, 1 μM Mg$^{2+}$ | 77.2 | 31.5 |
| | WT | LPM | 88.9 | 36.3 |
| | *ΔacrAB* | LB | 31.9 | 13.0 |
| *E. coli* (K12) | WT | LB | >150 | >61.0 |
| | *lptD4213* | LB | 25.9 | 16.0 |
| | *ΔtolC* | LB | 20.1 | 8.2 |

1 1X MIC is defined as MIC$_{95}$

cultures exposed to JD1 for CFU enumeration. Within 15 minutes of treatment with 2x MIC JD1, CFU recovery declined 100-fold for *S.* Typhimurium in LB with PMB and for the *E. coli lptD4213* mutant strain in LB (Fig 2C, 2D, 2G and 2H). These data indicate that concentrations of JD1 above 1x MIC are bactericidal. JD1 also inhibited the growth and survival of lag-phase bacteria but not of early stationary phase bacteria (S2B–S2G Fig). The results of the growth and kill curves together suggest that disruption of the outer membrane potentates JD1. Moreover, the data reveal dose- and time- dependent conditions under which responses to JD1 treatment can be unraveled.

## The AcrAB-TolC efflux pump protects bacteria from JD1

For virulence, *S.* Typhimurium requires the efflux pump AcrAB-TolC, a member of the RND (resistance-nodulation-cell division) family [40,41]. To establish whether JD1 may be expelled by efflux pumps, we used a Hoechst 33342 accumulation assay [42]. Wild-type *S.* Typhimurium treated with 21 μM of JD1 retained 26 +/- 0.4% (mean +/- standard deviation) more Hoechst than DMSO-treated bacteria, suggesting JD1 may compete with Hoechst for export from the cell. We also noted that *S.* Typhimurium and *E. coli* lacking *acrAB* or *tolC*, respectively, were more sensitive to JD1 than to the wild-type parent strains (Fig 2A, 2B, 2E, 2F, 2I and 2J and Table 1). The AcrAB-TolC efflux pump may therefore protect bacteria from the effects of JD1.

AcrB spans the bacterial inner membrane and captures substrates for export from the cell. We therefore used isothermal calorimetry (ITC) with purified AcrB to determine whether JD1 may bind AcrB (Fig 3A and 3B) [43]. The JD1 equilibrium dissociation constant (K$_D$) was 0.52 μM, suggesting binding. Since the binding enthalpy (ΔH) is favorable (-14.3 kcal/mol) at 25˚C, the interaction between JD1 and AcrB may involve hydrogen bonding and hydrophobic moieties. The unfavorable change in entropy (ΔS) (-19.3 kcal/mol) suggests the involvement of conformational changes. The binding affinity of JD1 for AcrB is similar to that of EPM30,

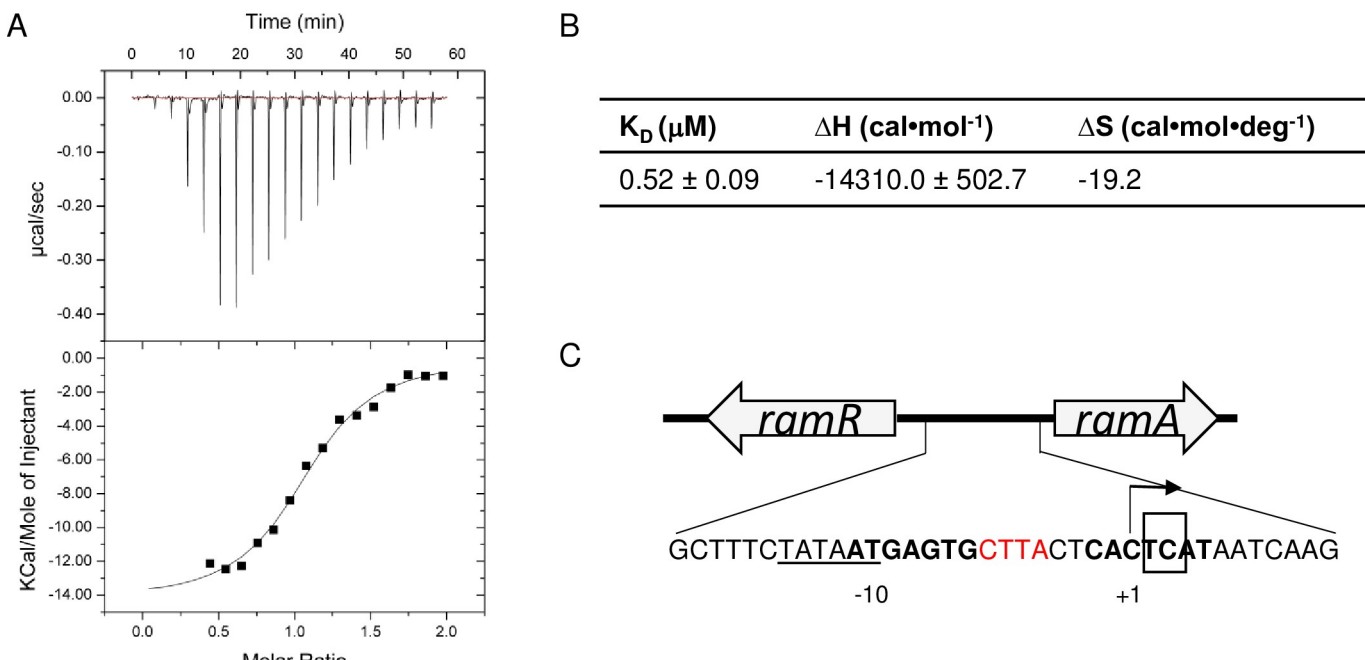

**Fig 3. JD1 appears to be a substrate for the AcrAB-TolC efflux pump.** A) Representative ITC for the binding of JD1 to *E. coli* AcrB. Each peak in the upper panel corresponds to the injection of 2 µL of 100 µM of JD1 in buffer containing 20 mM Na-HEPES (pH7.5), 0.05% DDM and 5% DMSO into the reaction containing 10 µM of *E. coli* monomeric AcrB in the same buffer. The lower panel shows the cumulative heat of reaction displayed as a function of injection number. The solid line is the least-square fit to the experimental data. B) $K_d$, enthalpy and entropy of the JD1-AcrB interaction. C) Diagram showing the *ramR* (*ramA* repressor) and *ramA* loci. Bold areas denote where the RamR homodimer binds to repress *ramA* expression. Base pairs in red are missing in all six JD1-resistant mutant strains. The box indicates the base pair deletion in BN10055 that interferes with RamR binding and increases efflux [47].

EPM35 and EPM43 for AcrB, as measured using ITC [21]. In addition, the dissociation constants are in good agreement with data from ITC measurements for other RND transporters; the *Campylobacter jejuni* CmeB and *Burkholderia multivorans* HpnN transporters interact with their corresponding substrates in the micromolar range [44,45]. These data indicate that JD1 binds to and may be a substrate for AcrAB-TolC.

## *S.* Typhimurium can develop resistance to JD1 by increasing AcrAB-TolC efflux pump activity

To determine how bacteria may become resistant to JD1, we examined the genotypes and phenotypes of six strains that we independently evolved in the presence of JD1. Whole-genome sequencing revealed that all six resistant strains had the same four-base-pair deletion within the promoter region of *ramA* (Fig 3C), which encodes a transcriptional activator of *acrAB* [46–50]. The 4 base-pair deletion overlaps with the site in which the RamR repressor binds to block *ramA* transcription. In an *S.* Typhimurium clinical isolate with a 2 base-pair (TC) deletion just downstream of our 4 base-pair deletion, RamR binding is reduced and the expression of *acrB* is increased by at least 3-fold [48].

To assay the efflux phenotypes of the mutant strains we exploited the fact that the AcrAB-TolC efflux pump is minimally active in the absence of glucose [51]. Without glucose, the resistant mutants accumulated as little or less of the AcrAB-TolC substrate Nile red than the Mar1 control strain (Table 2, column 2), which has enhanced AcrAB-TolC activity [42]. The presence of the protonophore carbonyl cyanide m-chlorophenyl hydrazone (CCCP) during glucose depletion increased Nile red accumulation in all six mutant strains more than in WT and

**Table 2. Nile red accumulation and loss in JD1-resistant mutants.**

| Strain | Nile red accumulation at Time 0 (no glucose) | | | Slope of Nile red loss after CCCP removal[1] | |
|---|---|---|---|---|---|
| | - CCCP[2] | + CCCP | Ratio, +/- CCCP | - glucose | + glucose |
| **Wild type** | 1.00 | 2.27 | 2.3 | 1 | 8.0 |
| **Mar1** | 0.52 | 2.09 | 4.0 | 1.1 | 9.4 |
| **Mutant 6** | 0.51 | 4.76 | 9.3 | 2.0 | 23.6 |
| **Mutant 5** | 0.47 | 2.24 | 4.8 | 1.5 | 11.4 |
| **Mutant 3** | 0.39 | 4.82 | 12.3 | 2.7 | 24.8 |
| **Mutant 2** | 0.37 | 4.35 | 11.9 | 3.1 | 21.4 |
| **Mutant 4** | 0.25 | 1.76 | 7.1 | 1.3 | 9.1 |
| **Mutant 1** | 0.04 | 0.25 | 7.0 | 0.24 | 0.8 |

1 Slope was calculated for the linear range of the experiment (the first 30 minutes) and normalized to wild type without glucose.

2 Mutants are arranged from highest to lowest Nile red retention in the absence of CCCP.

the Mar1 strain (Table 2 columns 3, 4). Mutant strain 1 accumulated very little Nile red with or without CCCP, suggesting it exports the dye so quickly that no accumulation is possible. Both mutant strains 1 and 4 also acquired mutations in the *cstAb* gene, which encodes a predicted carbon starvation protein. Mutant 1 had a deletion of four codons (encoding L123, A124, G125, V126) followed by a single nucleotide deletion that caused a frameshift at V127. Mutant 4 had a double nucleotide deletion that caused a frameshift at A124, a deletion of two codons (encoding G125 and V126), and a single nucleotide deletion causing a frameshift at V127. Whether carbon starvation contributes to Nile red efflux and/or exclusion, an alternative hypothesis, has not been explored. Mutant strains 2, 3 and 6 had especially high ratios of Nile red +/- CCCP, suggesting that these isolates have particularly active efflux pumps. Upon removal of CCCP and the addition of glucose, all of the mutants increased their rates of Nile red export, indicating that the pumps remained, as expected, dependent on proton motive force (Table 2, columns 5 and 6). These data support the conclusion that the six independently isolated mutants have increased efflux pump activity, and that bacteria can develop resistance to JD1 by increasing the rate of efflux and possibly exporting the compound.

## JD1 damages bacterial cell membranes if the outer membrane barrier is disabled

JD1 is highly lipophilic molecule with a calculated LogP of 12.5; LogP increases with lipophilicity, as it reflects the ability of a molecule to partition in octanol compared to water. JD1 may therefore integrate into membranes. To determine whether the compound damages bacterial membranes we used the fluorescent probe 3,3'-dipropylthiadicarbocyanine iodide [DiSC$_3$(5)], which accumulates in the lipid bilayer and becomes quenched. DiSC$_3$(5) fluorescence increases upon release of the probe from compromised membranes [52]. As anticipated, treatment of bacteria with gramicidin, a mixture of large (1,882 Da), pore-forming peptides [53] that depolarizes membranes, increased DiSC$_3$(5) fluorescence (Fig 4A). The addition of JD1 to *S.* Typhimurium in LB with PMB or to the *lptD4213* strain in LB also rapidly increased DiSC$_3$(5) fluorescence (Fig 4A and 4B). In contrast, the Δ*tolC* mutant strain was resistant to gramicidin and had a delayed response to JD1 (Fig 4C). To determine whether differences in sensitivity to JD1 could reflect differing degrees of outer membrane permeability, we compared nitrocefin access to the periplasm across strains and conditions. The *E. coli lptD4213* mutant strain in LB saturated nitrocefin hydrolysis within minutes (Fig 4D). *S.* Typhimurium in LB with PMB

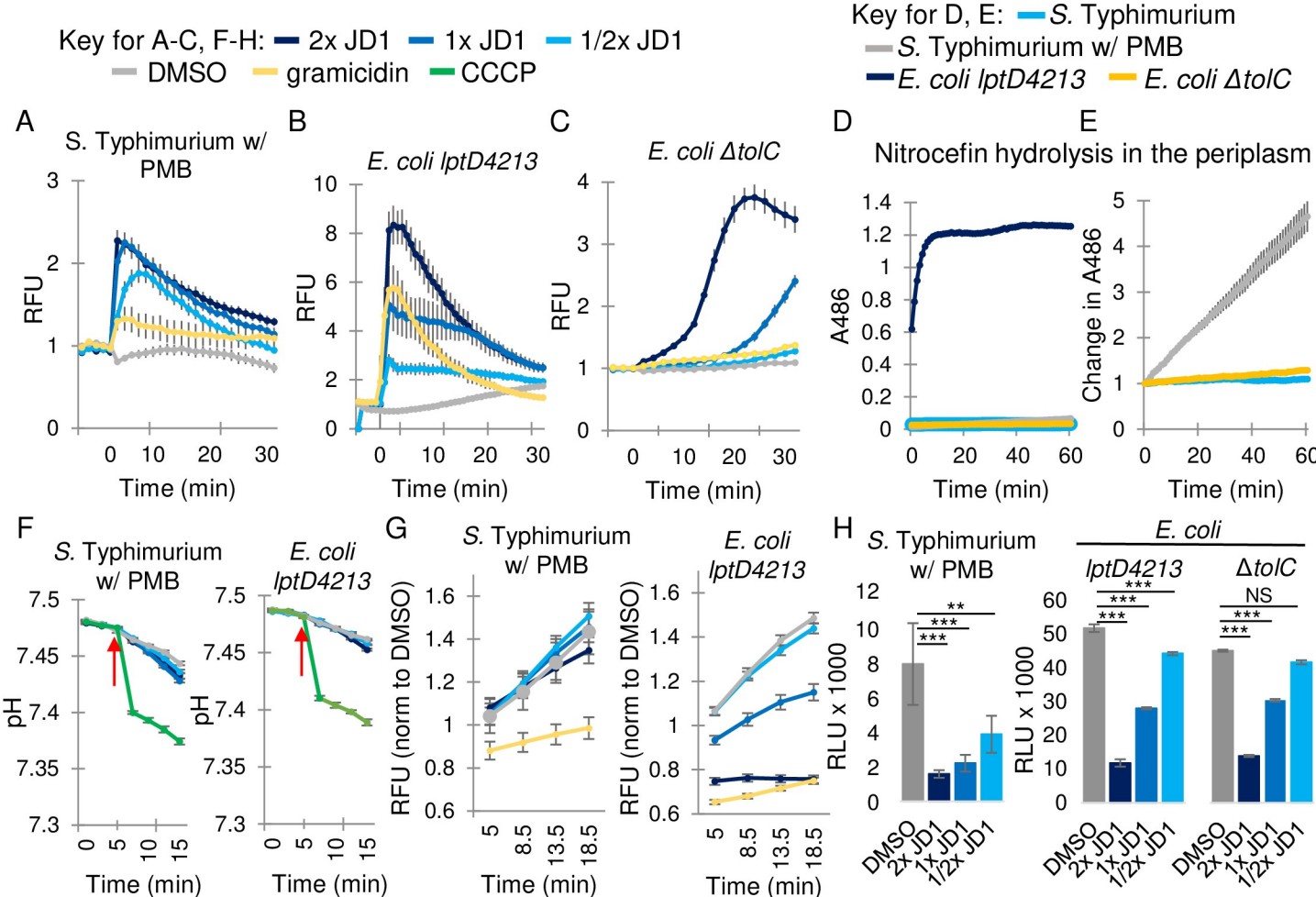

**Fig 4. JD1 damages the bacterial cytoplasmic membrane without disrupting the pH gradient or respiration.** A) Cell membrane potential was monitored with the fluorescent dye DiSC$_3$(5) for A) *S.* Typhimurium in LB with 0.5 µg/mL PMB, and B, C) the *E. coli lptD4213* and *ΔtolC* mutant strains in LB. Cells were treated at time 0 with DMSO, gramicidin (32 µg/mL), or the corresponding MIC$_{95}$ concentration of JD1 (Table 1). Average and SEM of three biological replicates performed with technical triplicates are shown for S. Typhimurium and *E. coli ΔtolC*. Average and SD from a representative of four biological replicates performed with technical triplicates are shown for *E. coli lptD4213*. Data are normalized to DMSO. D, E) Outer membrane permeability was monitored based on nitrocefin hydrolysis in the periplasm for *S.* Typhimurium in LB with 0.5 µg/mL PMB, and the *E. coli lptD4213* and *ΔtolC* mutant strains in LB. E) The graph in D without *E. coli lptD4213*, shown as a change in fluorescent signal. Average and SEM of three biological replicates performed with technical triplicates. F) Intracellular pH was monitored with the fluorescent probe BCECF in cells grown as in A and B and then treated with the indicated concentrations of the protonophore CCCP (1 mM), DMSO (vehicle), or JD1 at the time shown by the red arrow. Average and SEM of three biological replicates performed with technical triplicates. G) Respiration rates of strains grown as in A and B, incubated with resazurin, and treated at time 0 as indicated. Average and SEM of three biological replicates performed with technical triplicates and normalized to DMSO at time 0. H) Intracellular ATP levels measured using the Promega BacTiter-Glo kit for cells grown as in A-C after 15 minutes of treatment with DMSO or JD1. Average and SEM of three biological replicates performed with technical triplicates. $^{**}$ $P \leq 0.005$, $^{***}$ $P \leq 0.001$.

gradually and modestly hydrolyzed nitrocefin (Fig 4E). The *E. coli ΔtolC* mutant strain in LB was completely insensitive to nitrocefin, consistent with resistance to gramicidin and the delayed effect of JD1. These data indicate that the outer membrane of the *ΔtolC* strain remained intact, and that JD1 damages bacterial cell membranes in proportion to its ability to cross the outer membrane and/or avoid export.

## JD1 has little effect on the pH gradient and respiration but reduces ATP

If JD1 damages membranes, it may disrupt the membrane pH gradient and/or respiration. However, monitoring of intracellular pH [54] showed no effect of JD1 treatment in any of the strains or

conditions tested (Fig 4F). Similarly, the cellular reduction potential [55] for *S.* Typhimurium in LB with PMB was unaffected by JD1 (Fig 4G). In the *E. coli lptD4213* mutant strain in LB, high doses of JD1 inhibited respiration (Fig 4G), potentially due to the considerably higher outer membrane permeability of this strain (Fig 4D). Since respiration and the pH gradient contribute to ATP production, we monitored ATP accumulation. JD1 decreased ATP accumulation in all three strains within 15 minutes of treatment (Fig 4H). In summary, the dissipation of ATP without significant effects of JD1 on the pH gradient or respiration, combined with the rapid and strong release of $DiSC_3(5)$ from the membrane, suggest that the primary target of JD1 is the bacterial cell membrane.

## Inner membranes become rapidly permeable and more fluid upon JD1 treatment

One way to damage a membrane is to physically disrupt its barrier function. We therefore established whether JD1 treatment allows propidium iodide (PI), a cell impermeable dye, to breach the inner membrane, bind DNA, and fluoresce. Exposure to JD1 at 2X MIC significantly increased PI fluorescence within 10 minutes, compared to the DMSO control (Fig 5A), indicating that JD1 disrupts the barrier function of the cell membrane.

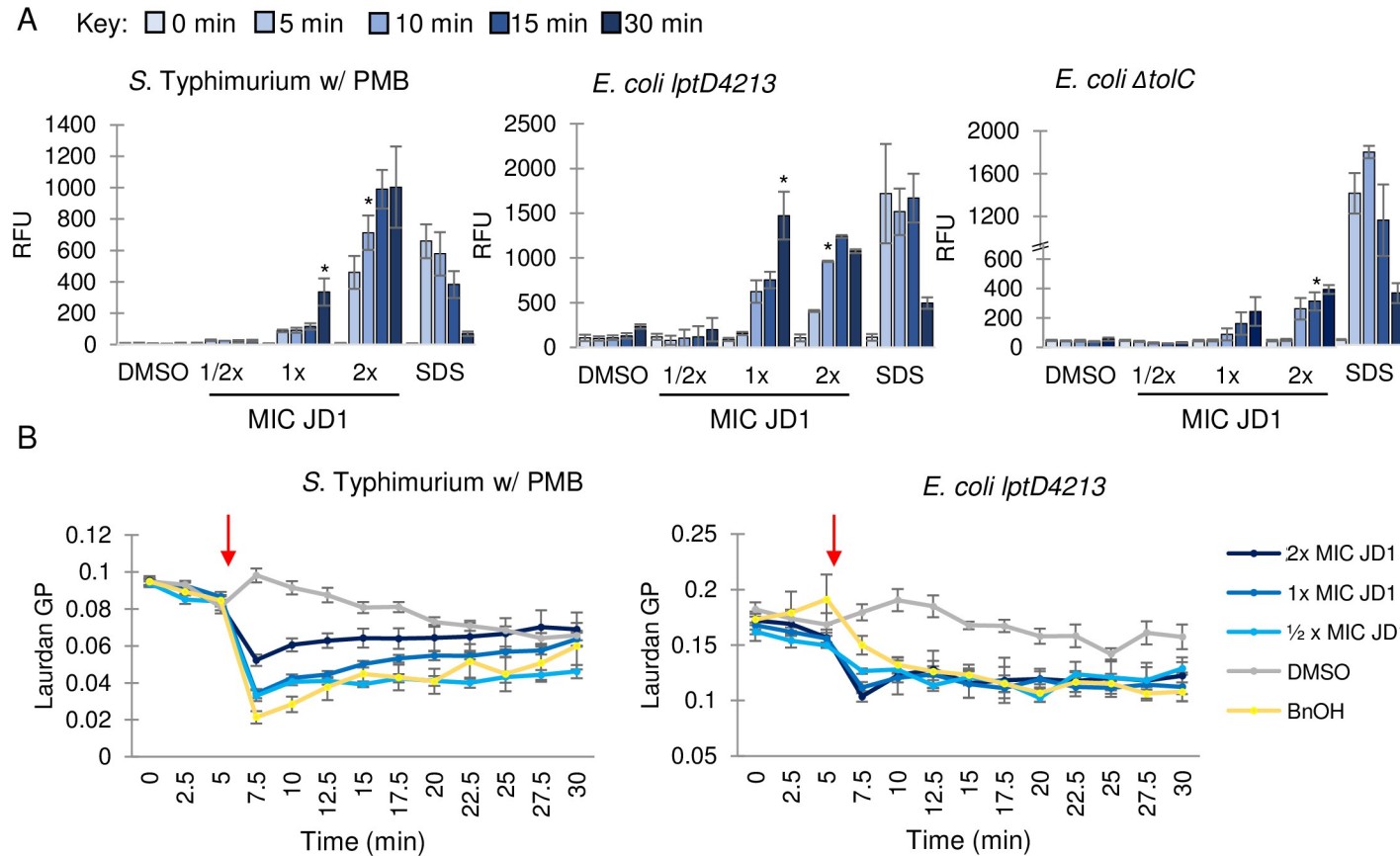

**Fig 5. JD1 perturbs membrane barrier function and fluidizes membranes.** A) Cell membrane permeability was monitored by PI fluorescence for *S.* Typhimurium in LB with 0.5 μg/mL PMB, and the *E. coli lptD4213* and *ΔtolC* mutant strains in LB. Cells were treated at time 0 with the corresponding $MIC_{95}$ concentration of JD1 (Table 1) or with DMSO or 0.008% SDS and samples were processed at the timepoints shown. Average and SEM of three biological replicates performed with technical triplicates. Asterisks indicate the first time point of JD1 treatment that resulted in a significant increase in PI fluorescence; all time points after the asterisk were also significant. * $P \leq 0.05$ as determined by ANOVA. B) Membrane fluidity as monitored by laurdan generalized polarization (GP) for *S.* Typhimurium in LB with 0.5 μg/mL PMB, and the *E. coli lptD4213* mutant strain in LB. Cells were treated at the time indicated (red arrow) with DMSO, benzyl alcohol (BnOH) (50 mM), or the corresponding $MIC_{95}$ concentration of JD1 (Table 1). Average and SEM of three biological replicates performed with technical triplicates.

Membrane barrier function can be compromised by an increase or decrease in membrane fluidity. Laurdan is a hydrophobic chemical that integrates into membranes, and its emission spectrum (reported as generalized polarity (GP)) is sensitive to changes in membrane fluidity [56]. We loaded cells with Laurdan and exposed them to either DMSO, the membrane fluidizer benzyl alcohol [57], or JD1. Benzyl alcohol and JD1 treatment rapidly increased fluidity in *S.* Typhimurium in LB with PMB and the *E. coli lptD4213* mutant strain in LB (Fig 5B). We found that Laurdan was toxic to the *E. coli ΔtolC* mutant strain at concentrations high enough to differentiate signal from noise, which may suggest that laurdan is an efflux pump substrate. From these data we conclude that treatment with JD1 causes a rapid increase in membrane fluidity and enables PI to leak through the inner membrane barrier.

## Microscopy reveals JD1-induced bacterial inner membrane distortions

Disruptions in membrane barrier function and fluidity may reflect structural changes visible by microscopy. We therefore used the lipophilic dye Nile red, which fluoresces upon integration into cell membranes, in combination with super-resolution structured illumination microscopy (SR-SIM) to examine JD1-treated cells. Live imaging was performed with stained bacteria on agar pads containing 1x MIC JD1. Since polymyxins do not reliably diffuse through agar [58] these experiments were performed with the *E. coli lptD4213* (Fig 6A) and *ΔtolC* (Fig 6B) mutant strains in LB, and not the wild-type *S.* Typhimurium in LB with PMB. Within the four to seven minutes required to focus the microscope, membrane distortions were apparent in cells on pads containing JD1 compared to DMSO. On the outside of the bacteria, wisps or circles of Nile red stained membrane appeared. Within cells, bright Nile red puncta formed, followed by patches or circles that grew in size. These data indicate that given a permeable outer membrane or loss of efflux pumps, JD1 rapidly causes membrane distortions to form.

## JD1 at high concentrations damages host cell and mitochondrial membranes

To characterize JD1 toxicity to eukaryotic cells we monitored macrophage membrane integrity with a standard lactate dehydrogenase release assay (LDH). JD1 had a half maximal cytotoxic concentration ($CC_{50}$) of 12 +/- 6.6 µM and 10 +/- 4.2 µM in uninfected and infected macrophages, respectively, and of 28 +/- 11 µM in HepG2 cells. To establish whether and when JD1 damages eukaryotic cell membranes, we loaded RAW 264.7 cells with the lipophilic dye Nile red, treated with DMSO or JD1, and imaged the live cells every 60 minutes for 16 hours (Fig 7A and S7–S15 Videos). Cells treated with as much as 8 µM of JD1 were alive and dividing at 16 hours. However, at all treatment concentrations, Nile red puncta, signifying regions of fluid lipids [59,60], accumulated in a dose- and time-dependent manner and grew in size over time. At 16 µM, treated cells shed pieces of membrane and developed abnormal morphologies. Cells treated with 32 µM JD1 developed Nile red puncta after two hours, and by five hours approximately half appeared to undergo cell death. These observations indicate that JD1 increases membrane fluidity in mammalian cells and at high concentrations is toxic.

Mitochondrial membranes are more similar in composition to bacterial cell membranes than to mammalian cytoplasmic membranes [61]. We therefore monitored the response of mitochondrial membrane polarization to cell treatment with JD1. The fluorescent dye tetramethyl rhodamine (TMRM) accumulates in the inner membrane and fluoresces in response to membrane potential. RAW 264.7 cells pre-loaded with TMRM were treated with compound or controls and imaged every 10 minutes for 80 minutes. CCCP treatment quickly decreased TMRM fluorescence, reflecting membrane depolarization (Fig 7B). Treatment with 20 or 40 µM of JD1 led to a rapid increase in fluorescence, suggesting hyperpolarization. These data

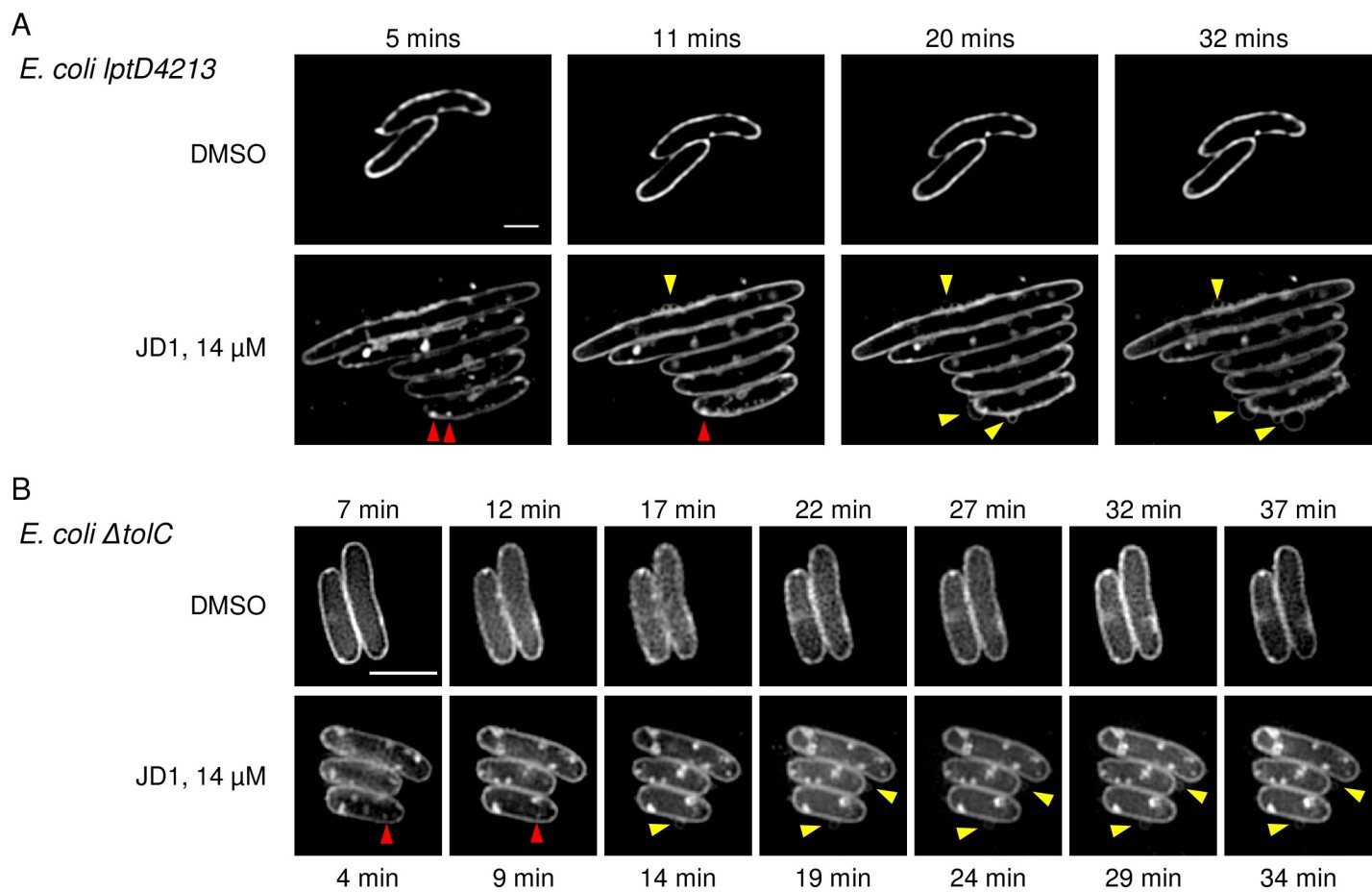

**Fig 6. Bacterial membranes become distorted in response to JD1 treatment.** Nile red staining and SR-SIM imaging of *E. coli* A) *lpt4213* and B) *ΔtolC* mutant strains on agar pads containing DMSO or JD1 at 14 μM. Cells were grown to mid-log phase, incubated with 30 μM of the lipophilic dye Nile red and placed on an agar pad. Imaging began as soon as cells were in focus (4–7 minutes) and continued for thirty minutes. Representative micrographs of three biological replicates are shown. Yellow arrowheads show Nile red wisps or circles outside of cells. Red arrowheads indicate puncta. Scale bar is 2 μm.

indicate that JD1 has modest effects on mitochondrial membrane potential at concentrations approximately 4-8-fold higher than are required to kill intracellular *S.* Typhimurium.

## JD1 reduces bacterial tissue colonization in mice

To establish whether JD1 decreases *S.* Typhimurium colonization of tissues in mice, we inoculated C57Bl/6 mice intraperitoneally with $8 \times 10^3$ wild-type bacteria and then treated with 1 mg/kg of JD1 intraperitoneally at 10 minutes and 24 hours post-infection. All mice that received JD1 survived in good condition out to 48 hours, at which time the spleen and liver were harvested. Enumeration of tissue CFU revealed that treatment with JD1 reduced *S.* Typhimurium colonization in the spleen ($P < 0.05$, Mann-Whitney; Fig 7C). There was not a significant difference in bacterial load in the livers, but the four livers from JD1-treated mice with the lowest CFU lacked visible abscesses, whereas all control-treated livers had abscesses. Thus, JD1 was tolerated *in vivo* and had antibacterial potency.

## Cholesterol protects neutral lipids from JD1

Finally, to determine whether JD1 may be more damaging to bacterial than to eukaryotic membranes, we examined synthetic vesicles loaded with the self-quenching fluorescent dye

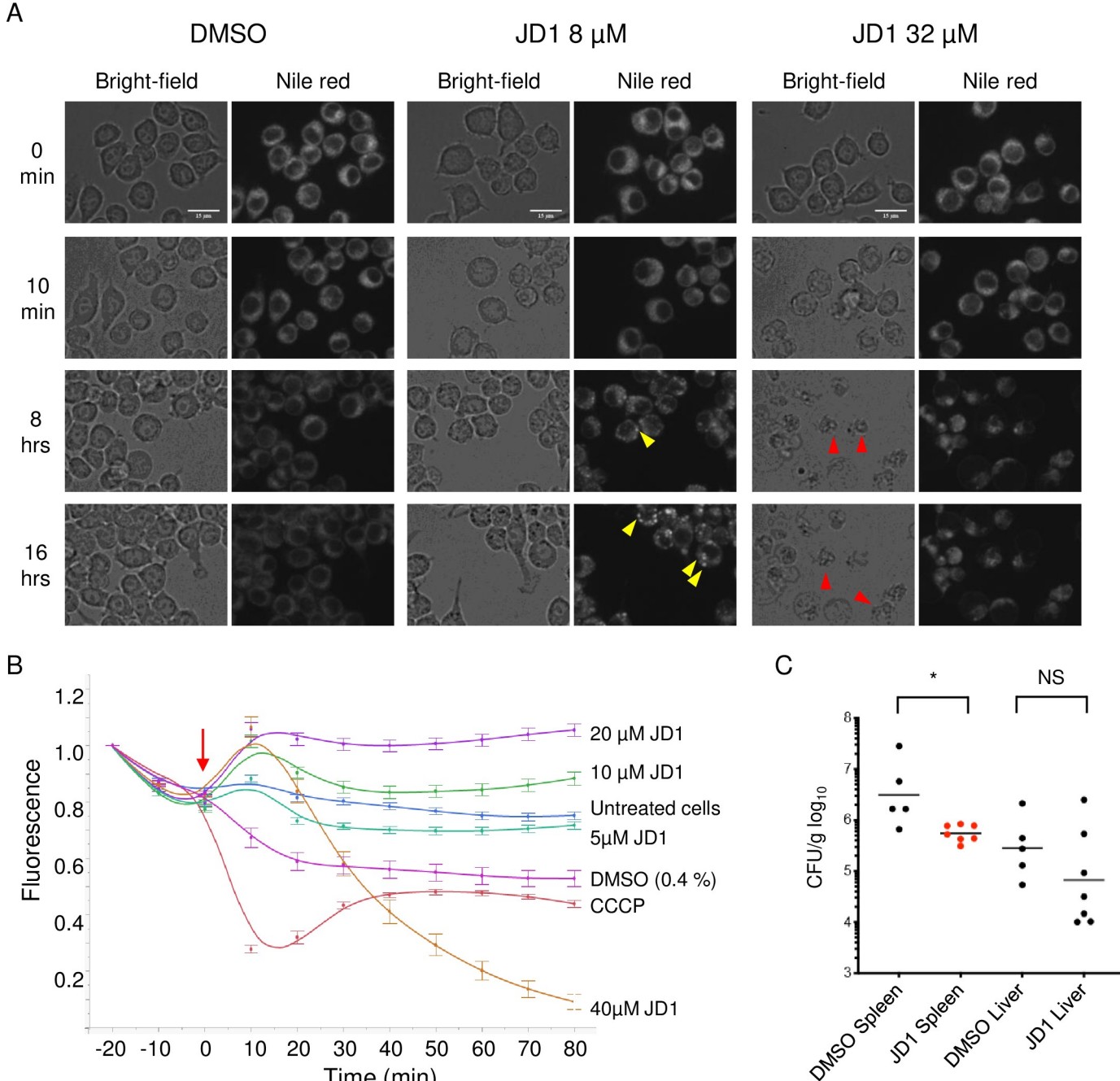

**Fig 7. JD1 is toxic to host cell membranes at high concentrations but has antimicrobial activity in mice.** A) RAW 264.7 macrophages were stained with Nile red then treated with DMSO or dilutions of JD1. Cells were imaged every 10 minutes for 16 hours and representative micrographs are shown. Yellow arrowheads indicate examples of Nile red puncta; red arrowheads indicate examples of dead cells. Scale bars are 15 μm. B) Macrophages were stained with the mitochondrial membrane potential indicator TMRM and were imaged every ten minutes for 80 minutes. Cells were treated (red arrow) with DMSO, CCCP, or dilutions of JD1. Averages and SEM of three biological replicates performed with technical triplicates and normalized to time 0. C) C57Bl/6 mice were intraperitoneally inoculated with 8 x 10³ *S.* Typhimurium CFU. At 10 minutes and 24 hours after infection, mice were dosed with 1 mg/kg of JD1 by intraperitoneal injection. Mice were euthanized 48 hours after infection. The spleen and liver were homogenized and plated for enumeration of CFU. * $P < 0.05$, Mann-Whitney.

sulforhodamine B. Leakage of sulforhodamine B leads to its dequenching and increases its fluorescence, offering a highly sensitive way to monitor membrane integrity [62]. We prepared liposomes with a lipid composition similar to Gram-negative bacterial cell membranes (67%

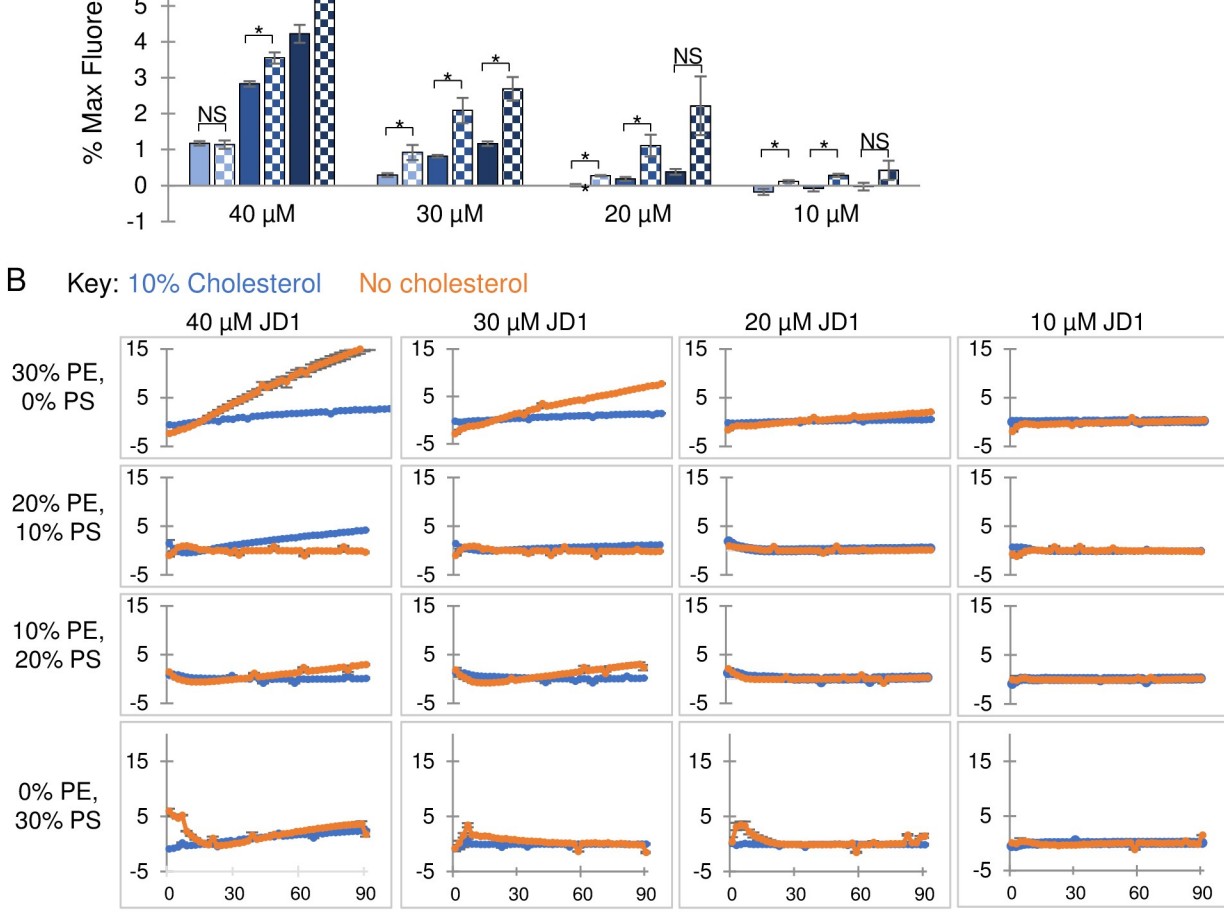

**Fig 8. In the presence of the neutral lipid PE, cholesterol protects liposomes from damage caused by JD1.** A) Liposomes with phospholipid content similar to mammalian (60% PC, 20% PE, 10% PS, 10% cholesterol) or *E. coli* (67% PC, 23% PG, and 10% CL) membranes were loaded with the fluorescent probe sulforhodamine B. Liposomes were treated with the indicated concentrations of JD1 or with control DMSO and monitored for fluorescence over a 90 minute period. Data are normalized to DMSO. Student's t-tests comparing the two indicated samples, * P ≤0.05, ** P ≤0.005. B) Liposomes composed of 60% or 70% PC with 10% or 0% cholesterol, respectively, and with PE and PS concentrations ranging from 0–30% or 30–0%, respectively were exposed to JD1 and examined for sulforhodamine B release. Averages and SEM of three biological replicates performed with technical triplicates.

phosphatidylethanolamine (PE), 23% phosphatidylglycerol (PG), and 10% cardiolipin (CL))
[63]. We also constructed liposomes similar to mammalian cell membranes, with 60% PC,
20% phosphatidylethanolamine (PE), 10% phosphatidylserine (PS), and 10% cholesterol [10].
Both types of liposomes were damaged by JD1 treatment, as indicated by increases in sulforho-
damine B fluorescence (Fig 8A). However, the mammalian-like membranes were consistently
more resistant to JD1 across dose and time than the bacterial-like membranes.

Bacterial cell membranes are substantially more negatively charged than mammalian cell
membranes, as negatively charged phospholipids (PG and CL) comprise approximately 33%

of the membrane as opposed to only 10% (mainly PS) in mammalian cells [10]. Alternatively, or in addition, the presence of cholesterol in the mammalian membrane may reduce the potency of JD1. We tested these hypotheses by constructing liposomes containing 30–0% PE and 0–30% PS, respectively, and with or without 10% cholesterol (Fig 8B). Liposomes containing both cholesterol and 30% PE (0% PS) were 6-fold more resistant to JD1 compared to their counterparts lacking cholesterol (Fig 8B). Reducing the PE:PS ratio eliminated cholesterol-mediated protection against JD1. These data indicate that JD1 acts directly on membranes, destabilizes them, and has a preference for bacterial-like membranes. Moreover, the results suggest that mammalian membranes are protected from JD1 by a combination of cholesterol and neutral lipids.

## Discussion

We used a fluorescence-based cell culture assay to identify compounds with non-traditional antimicrobial capabilities from chemical and drug libraries. This approach uncovered inhibitors of bacterial efflux pumps, a stimulator of macrophage autophagy and an antibiotic used against Mycobacterium species that was unexpectedly potent against Gram-negative bacteria during infection [21,22,25]. Within the current work, we examined the mechanism of action of a compound identified by SAFIRE, JD1, that strongly reduces *S.* Typhimurium replication in macrophages and disrupts the bacterial cell membrane upon breaching the outer membrane barrier.

### Bacterial resistance to JD1 –outer membranes and efflux pumps

JD1 does not have antibacterial activity in standard microbiological media but inhibits bacterial growth and survival under conditions that compromise the outer membrane. In particular, JD1 is potentiated by conditions that weaken the LPS layer, including the presence of the cAMPs or low concentrations of magnesium. The condition most effective at potentiating JD1 was the presence of PMB, a cAMP that vigorously damages the outer membrane [38]. PMB also generates hydroxyl radicals that oxidize proteins within the inner membrane [37,64]. However, we used PMB at concentrations that do not permeabilize the inner membrane [25] and found that iron chelation, which reduces hydroxyl radical production [65], did not reduce JD1 potentiation by PMB. It therefore appears that outer membrane permeability facilitates JD1 antimicrobial activity.

Efflux pumps also help to protect bacteria against JD1. In particular, the RND efflux pump AcrAB-TolC may capture JD1 and expel it from the cell: selection for resistance to JD1 yielded strains with mutations that increase efflux, and JD1 was observed to bind AcrB and may therefore be a substrate for export. We hypothesize that when the outer membrane LPS layer is intact, JD1 enters the periplasm at a low rate, such that export by AcrAB-TolC is sufficient to protect the bacteria. Upon damage to the outer membrane, JD1 entry into the periplasm may overwhelm efflux pumps, enabling the compound to accumulate and insert into the bacterial cell membrane.

### Mechanism of action of JD1 in Gram-negative bacteria

The primary target of JD1 in bacteria appears to be the bacterial cell membrane, as the compound rapidly and strongly depolarizes membranes and increases membrane fluidity. Moreover, within 5–10 minutes of exposure to JD1, the bacterial cell membrane becomes permeable, and membrane structural irregularities appear, including the accumulation of highly fluid membrane patches. Increased membrane fluidity heightens the permeability of membranes to many non-ionic substances [66] and is consistent with the observed

maintenance of respiration and the pH gradient in the presence of JD1. We speculate that when JD1 accesses the bacterial cell membrane, it intercalates, increasing membrane fluidity and permeability, and arrests growth.

We note that JD1 is considerably more efficacious against wild-type *S*. Typhimurium within macrophages and HeLa cells than in the media conditions or genetic backgrounds tested. This discrepancy does not appear to be due to increased access of gentamicin to bacteria due to minor host membrane damage caused by JD1 because we did not observe synergy between gentamicin and JD1 during infection. However, we cannot rule out that in mammalian cells, JD1 causes bacterial killing by an unknown mechanism(s) that could interfere with any bacterial or host process needed for bacterial replication or survival. Nevertheless, the simplest explanation for the higher JD1 antibacterial activity in cells compared to broth is that the phagosome is much more complex and dynamic. For example, the *acrAB* locus is required for *S*. Typhimurium growth in macrophages and mice but not in media [40,67], indicating that the phagosomal lumen is a unique environment. Moreover, the known bacterial necessity for efflux pumps in macrophages irrespective of JD1 [40,41], may contribute to *S*. Typhimurium sensitivity to JD1 during growth in macrophages. Thus, JD1 may be potent during infection because it exploits host damage to the LPS layer and because efflux pumps are occupied with the export of host antimicrobials [15,68–70].

## Neutral lipids and cholesterol protect membranes from JD1

JD1 is toxic to mammalian cells at concentrations significantly higher than are needed to kill S. Typhimurium in macrophages. Observations of toxicity include examination of fixed images from SAFIRE assays, live imaging of JD1-treated macrophages stained with a membrane dye or a marker of mitochondrial membrane potential, and LDH release from JD1-treated cells. Assays with liposomes revealed that JD1 has a preference for bacterial-like membranes, which contain more negatively charged lipids than mammalian plasma membranes. Moreover, as the percentage of neutral lipids in a liposome increased, cholesterol protected liposomes from JD1-induced damage. These observations suggest that mammalian membranes may be resistant to JD1 in part because they contain a higher proportion of neutral and fewer negatively charged phospholipids than Gram-negative bacteria [71], and additionally, due to the presence of cholesterol. Cholesterol is a component of mammalian cellular and organellar membranes, including those of mitochondria [72]. Since at physiological temperatures cholesterol reduces membrane fluidity [12], it may counteract the ability of JD1 to fluidize membranes and protect the neutral lipids in mammalian membranes from the activity of JD1.

## JD1 differs from current antibiotics and membrane-targeting compounds

There is only one FDA-approved antibiotic, daptomycin, that disrupts bacterial cell membranes as its mechanism of action. Daptomycin is a large (1,620 Da) lipopeptide that is only effective in Gram-positive bacteria. It is thought to integrate into membranes and increase their rigidity, causing the disassociation of essential peripheral membrane proteins [73]. While daptomycin may have difficulty crossing the Gram-negative bacterial outer membrane barrier, *E. coli* protoplasts are nevertheless resistant to daptomycin despite having an accessible inner membrane [11]. This resistance of *E. coli* protoplasts may reflect the lower proportion of negatively charged phospholipids in Gram-negative compared to Gram-positive cell membranes [11]. In particular, daptomycin appears to dock to the negatively charged lipid PG [74], which is more abundant in the membranes of Gram-positive bacteria [63]. Daptomycin, in summary, is much larger than JD1, is ineffective against Gram-negative bacterial membranes, and stiffens instead of loosens membranes.

Several additional lipophilic natural product compounds that disrupt bacterial cell membranes have been reported, and their mechanisms of action are instructive. Rhodomyrtone (443 Da) is derived from a flowering plant in the family Myrtaceae [75] and fluidizes the membranes of Gram-positive bacteria in broth culture [76]. In zebrafish embryos infected with *Streptococcus pneumoniae*, injection of two doses of 25 ng each of rhodomyrtone per embryo prevented bacterial accumulation [76]. Given an approximate embryo volume of 0.18 mL [77], the animals survived two dosages of 0.31 μM rhodomyrtone [76]. In murine experiments evaluating the anti-depressive effects of rhodomyrtone, dosages as high as 15 mg/kg (~34 μM) delivered intraperitoneally were tolerated [78]. However, cell culture experiments indicate that rhodomyrtone is toxic to multiple human cell lines at concentrations as low as 2 μM [79]. Another natural product, violacein (343 Da), permeabilizes the cell membranes of Gram-positive bacteria at concentrations as low as 4 μM. [80]. It also damages liposomes with lipid compositions similar to *E. coli* cell membranes [80]. Violacein appears to be moderately toxic to mammalian cells, killing 60% of HeLa cells at 6 μM, on the basis of Trypan blue staining [81]. Violacein has not to our knowledge been tested in a cell culture or animal model of infection. The continued study of rhodomyrtone and violacein as agents that damage membranes may reveal features of membrane biology germane to the development of antibacterials.

One lipophilic compound, metergoline, can damage the cell membrane of Gram-negative bacteria and inhibit infection of macrophages and mice [82]. However, this ergot-derived natural product has demonstrated effects on multiple eukaryotic cell receptors. Metergoline, has been examined for antidepressant activity in humans and is an antagonist of 5-HT$_{1B}$ serotonin receptors [83]. Regulators of 5-HT serotonin receptors modulate macrophage cytokines, mitigate infection with bacteria and fungi in macrophages and mice [21,84–86], and inhibit RND efflux pumps in *E. coli* [87,88]. In addition, metergoline inhibits sodium/potassium channels in *Xenopus laevis* and human cells [89–91]. These findings suggest that metergoline has pleiotropic effects in bacteria and eukaryotes.

### Lipophilic compounds and their utility for treating infections

Lipophilic compounds (LogP >2) tend to have low aqueous solubility and thus reduced bioavailability upon oral therapeutic delivery [92]. The oral delivery of lipophilic drugs in corn oil, for instance [93], enhances drug uptake across the small intestine [92]. For the treatment of serious infections, lipophilic drugs may have an advantage over hydrophilic drugs because the pathophysiologic changes that occur during severe inflammation favor the chemistry of hydrophobic antibiotics [94]. The distribution volume of hydrophilic drugs increases as leakiness of the vascular endothelium raises the volume of fluid in tissues, observed as edema. In this context, lipophilic drugs retain good tissue absorption. For instance, in septic patients, higher and increasingly toxic doses of hydrophilic antibiotics are typically required for effective treatment [94]. Clofazimine is a lipophilic (LogP 7.7) clinical antibiotic that damages membranes [95–98] and is used to treat infections with *Mycobacterium leprae* and *tuberculosis* [99–102]. Clofazimine is not active against Gram-negative bacteria in standard microbiological media [103–105] but is has efficacy in macrophages and mice against *S.* Typhimurium [25]. While JD1 is likely too lipophilic (LogP 12.5) to become a lead compound for antibiotic discovery, our observations demonstrate the feasibility of developing lipophilic antibiotics that target Gram-negative bacterial cell membranes and may ultimately have therapeutic value.

### Conclusions

The small molecule JD1 damages the cell membranes of Gram-negative bacteria under conditions in which the outer membrane is compromised and/or efflux pumps are occupied, as

occurs during infection. Mammalian cell membranes appear to be protected from JD1 by a combination of neutral lipids and cholesterol. JD1 was also able to mitigate infection in a mouse model without obvious whole-animal toxicity. These observations suggest the potential for small molecule antibiotics to exploit host damage to the outer membrane to reach and target the Gram-negative bacterial inner membrane. Such molecules would represent a new way to address the threat of resistance to existing clinical antibiotics.

## Methods

### Ethics statement

This study was carried out in accordance with the recommendations in the *Guide for the Care and Use of Laboratory Animals* of the National Institutes of Health. Protocols were approved by the University of Colorado Institutional Committees for Biosafety and Animal Care (2445). Euthanasia method: carbon dioxide asphyxiation.

### Media and reagents

Unless otherwise stated, bacteria were grown in lysogeny broth (LB) at 37˚C [106,107]. Where indicated, *S*. Typhimurium (SL1344 [108] or ATCC 14028) was grown in M9-lowMg (42 mM $Na_2HPO_4$, 22 mM $KH_2PO_4$, 18.7 mM $NH_4Cl$, 8.5 mM NaCl, 0.1% casamino acids, 1 μM $MgSO_4$, 2% glucose), or LPM (5 mM KCl, 7.5 mM NH4SO4, 0.5 mM K2SO4, 337μM KH2PO4, 8 μM MgCl2, 0.3% glycerol, 0.1% Casamino acids, 0.2% histidine, 80 mM MES, pH 5.0). Additives to media, where indicated, include 0.5 μg/mL PMB (Sigma-Aldrich), or 10 μg/mL PMBN (Sigma-Aldrich), or DFS at stated concentrations (Adooq Biosciences).

### SAFIRE and CFU assays

RAW 264.7 macrophages ($5 \times 10^4$ macrophages in 100 μL of complete DMEM) were seeded in 96-well black, glass-bottomed plates (Brooks Life Sciences) [21]. For experiments performed with HeLas, $1 \times 10^4$ cells were seeded. Twenty-four hours later, bacteria grown overnight in LB and diluted into 50 μL PBS were added to a final concentration of $1 \times 10^7$ CFU/mL, an approximate multiplicity of infection (MOI) of 30 bacteria to one cell. Forty-five minutes after bacterial addition, gentamicin (Sigma-Aldrich) was added to a final concentration of 40 μg/mL. Two hours after infection, 1 μL of compound (JD1; BTB12794; Molport) or vehicle control was added to the stated final concentration. For SAFIRE and CFU experiments in which the concentration of gentamicin was reduced from 40 to 10 μg/mL, this transition occurred two hours after infection, coinciding with compound treatment. At 17.5 hours post-infection, PBS containing MitoTracker Red CMXRos (Life Technologies) was added to a final concentration of 100 nM. At 18 hours after infection, 16% paraformaldehyde was added to a final concentration of 4% and incubated at room temperature for 15 minutes. Cells were washed, stained with 1 μM DAPI and stored in 90% glycerol in PBS until imaging. After 16 hours of treatment, samples were imaged on a spinning disk confocal microscope, and a MATLAB algorithm calculated bacterial accumulation (GFP fluorescence) within macrophages, as defined by DAPI (DNA) and MitoTracker Red, a vital dye for mitochondrial voltage. GFP+ macrophage area is defined as the number of GFP-positive pixels per macrophage divided by the total number of pixels per macrophage, averaged across all macrophages in the field [25]. For CFU determination, infections were performed as above except cells were seeded in 96-well tissue culture plates (Greiner). At 18 hours post-infection, wells were washed three times in PBS, lysed with 30 μL 0.1% Triton X-100, diluted, and plated to determine CFU.

## Live cell infection microscopy

Experiments were performed with RAW 264.7 cells between passages one and six. Cells were grown in complete DMEM to approximately 70–90% confluency. Cells were scraped, washed, resuspended and diluted to a final concentration of $5x10^5$ cells/mL. Cells (100 μL) were transferred to a 96-well black, glass-bottomed plate (Brooks Life Sciences) and incubated for 24 hours at 37˚C with 5% $CO_2$. Cells were infected with SL1344 *sifB*::GFP [109,110] at an MOI of 30. After 45 minutes the medium was exchanged for complete FluoroBrite DMEM (Thermo Fisher) with 40 μg/mL of gentamicin and incubated for an additional 45 minutes. Compounds were added as 50 μL aliquots to obtain the desired concentration in a final volume of 200 μL. After drug addition, the concentrations of gentamicin and DMSO were 30 ug/mL and 0.4%, respectively. Cells were time-lapse imaged on a Yokogawa CellVoyager CV1000 Confocal Scanner System with a 40x/0.6NA objective in an environmentally controlled chamber for 17 hours with images acquired every 30 minutes. Six fields per well were imaged with each field comprising five images sampled over a z-dimension of 15 μm. The resulting images were converted into maximum intensity projections and the GFP foreground signal from each well was extracted via a MATLAB R2018a (MathWorks) script and analyzed with JMP statistical software. The integrated density is defined as foreground signal obtained from maximum intensity projections of GFP$^+$ Macrophage Area across six microscope fields.

## Minimum inhibitory concentration determination

Overnight cultures were grown in LB or in M9-lowMg for samples tested in this medium. Cultures were diluted to an optical density at 600 nm ($OD_{600}$) of 0.01 in their respective testing media and distributed in 96-well flat-bottom plates. Compound was added to the desired final concentration, and the final DMSO concentration never exceeded 2%. Plates were grown at 37˚C with shaking and $OD_{600}$ was monitored (BioTek Synergy H1 or BioTek Eon). MICs were defined as the concentration at which 95% of growth was inhibited ($OD_{600}$).

## Growth curves and kill curves

Overnight cultures were diluted 1:100 in LB or LB with PMB and incubated at 37˚C until they reached mid-log phase ($OD_{600}$ 0.4–0.6). Samples were taken at time 0 and then compound or vehicle control was added. Cultures were incubated at 37˚C with agitation. At the time intervals indicated, aliquots were monitored for $OD_{600}$ and plated for CFU enumeration. Data for $OD_{600}$ and CFU/mL were normalized to time 0.

## Hoechst 33342 accumulation

As previously described [21], overnight *S*. Typhimurium cultures were washed three times in PBS and diluted to an $OD_{600}$ of 0.1 in PBS with 2.5 μM Hoechst 33342 in the presence of compound or vehicle control. Fluorescence was monitored on a BioTek Synergy H1 with a 360/40 nm excitation filter and 460/40 nm emission filter. Maximum fluorescence over 60 minutes was normalized to the signal from the equivalent number of heat-killed bacteria and had the background signal removed. The background signal was determined by subtracting the autofluorescence of compound incubated without bacteria prior to normalization to heat-killed bacteria.

## Isothermal Calorimetry (ITC)

AcrB protein was purified as described [43]. Briefly, the AcrB protein contains a 4·His tag at the C-terminus and was overproduced in *E. coli* BL21-Gold (DE3) cells (Stratagene) using the

plasmid derived from pSPORT1 (Invitrogen) [111]. Cells were grown in 6 L of LB medium with 100 lg/ml ampicillin and disrupted with a French pressure cell. The membrane fraction was collected and washed twice with buffer containing 20 mM sodium phosphate (pH 7.2), 2 M KCl, 10% glycerol, 1 mM EDTA and 1 mM phenylmethanesulfonyl fluoride (PMSF), and once with 20 mM HEPES–NaOH buffer (pH 7.5) containing 1 mM PMSF. The membrane proteins were then solubilized in 1% (w/v) n-dodecylb-D-maltoside (DDM). Insoluble material was removed by ultracentrifugation at 370,000 x g. The extracted protein was purified with $Cu^{2+}$-affinity and G-200 sizing columns [112,113]. The purified AcrB protein was then concentrated to a final monomeric concentration of 10 µM in buffer containing 20 mM Na-HEPES (pH 7.5) in 0.05% DDM. Similar protein purification procedures have been used to elucidate structure-function of RND transporters, including AcrB [43,113,114], CusA [115,116], MtrD [117,118], CmeB [44], AdeB [119], HpnN [45] and MmpL3 [120]. We also used these protein purification protocols for *in vitro* substrate transport study via the CusA transporter [116] and *in vitro* functional dynamics measurement of the CmeB transporter [44], indicating that these purified membrane proteins are fully functional *in vitro*.

Measurements were performed on a Microcal iTC200 (Malvern Panalytical) at 25˚C. Before titration, the protein was dialyzed against buffer containing 20 mM Na-HEPES (pH 7.5), 0.05% n-dodecyl-µ-maltoside (DDM) and 5% DMSO [21]. The Bradford assay was used to quantify protein concentration, which was adjusted to a final monomeric concentration of 10 µM. Ligand solution consisting of 100 µM JD1 in the aforementioned buffer was prepared as the titrant. Both the protein and ligand samples were degassed before loading the samples. Two µL injections of the ligand were used for data collection. Injections occurred at intervals of 60 seconds and lasted for 4 seconds. Heat transfer (µcal/s) were measured as a function of elapsed time (s). The mean enthalpies measured from injection of the ligand in the buffer were subtracted from raw titration data before data analysis with ORIGIN software (MicroCal). Titration curves fitted with a non-linear regression fitting to the binding isotherm provided the equilibrium binding constant ($K_A = 1/K_D$) and enthalpy of binding ($\Delta H$). Based on the values of $K_A$, the change in free energy ($\Delta G$) and entropy ($\Delta S$) were calculated with the equation $\Delta G = -RT \ln K_A = \Delta H - T\Delta S$, where $T$ is 2/3 K and $R$ is 1.9872 cal/K per mol. Calorimetry trials were also carried out in the absence of AcrB using the same experimental conditions. No change in heat was observed in the injections throughout the experiment.

## Evolution of resistant mutants and genetic analysis

To ensure that all isolates started with the same genetic background, a single colony of wild-type *S.* Typhimurium was resuspended and then distributed into six independent M9 low magnesium broth cultures containing 0.25x MIC of JD1. Each day growth was visible, cultures were diluted 1:100 in fresh medium containing 0.25x MIC additional JD1 until growth at 3x MIC was achieved (~12 passages). Isolates were recovered on LB agar and tested for heritable resistance with 3x MIC JD1. Genomic DNA from overnight LB cultures of resistant mutants and a parental control strain was extracted with the E.Z.N.A bacterial DNA kit (Omega Biotek). Library preparation and sequencing (MiSeq V2 2x150 paired end) was performed by the BioFrontiers Sequencing Facility at the University of Colorado Boulder. Data were analyzed for mutations using Snippy (https://github.com/tseemann/snippy).

## Nile red efflux assays

Nile red (Sigma-Aldrich) efflux assays were performed as previously described with slight modifications [21]. Briefly, an overnight culture was washed and re-suspended in PBS containing 1 mM $MgCl_2$ at an $OD_{600}$ of 2.0 in a glass tube. Cells were incubated at room temperature

for 15 minutes. CCCP was added (10 μM final; Sigma-Aldrich) and cells were incubated for another 10 minutes at room temperature. Nile red was added (10 μM final) and cells were incubated for 2 hours and 15 minutes at 37˚C in a roller drum. Samples were transferred to a one hour standing incubation at room temperature. Cells were pelleted at 10,000 $x\,g$ for 1 minute and resuspended in 200 uL PBS with 1 mM $MgCl_2$. Cells were transferred to a black 96-well plate (Greiner) and monitored (ex540/em650 nm) using a BioTek Synergy H1 plate reader. To stimulate efflux, glucose was added to a final concentration of 2 mM after the first read. Fluorescence was recorded every 50 seconds for 12 minutes. To determine the rate of efflux, the slope was calculated over the linear portion of the reaction, the first 2 minutes.

## Membrane potential assays

Membrane potential was measured using the potentiometric fluorescent probe $DiSC_3(5)$ (Invitrogen). Overnight cultures were subcultured 1:50 in fresh LB and grown to log phase ($OD_{600}$ 0.4–0.5) and then diluted to an $OD_{600}$ of 0.4. $DiSC_3(5)$ was added to a final concentration of 2 μM and the culture was incubated at 37˚C in a rotator for 15 minutes. Cells were captured on a 0.45 μm Metricel membrane filter (Pall) and resuspended in fresh LB. Resuspended cells (200 μL) were distributed into wells of a black 96-well plate (Greiner). Plates were monitored (ex650/em680 nm) on a BioTek Synergy H1 plate reader. After baseline fluorescence was recorded, compound was added to the desired final concentration and measurements were recorded for an additional 30 minutes.

## Nitrocefin hydrolysis assays

Nitrocefin hydrolysis assays were performed as previously described with slight modifications [121]. Overnight cultures of cells harboring the *p*FBH1 plasmid containing an ampicillin resistance gene were subcultured 1:50 followed by regrowth to mid-log phase (~2 hours). Cells were washed (20 mM $KPO_4$, pH 7.0, 1 mM $MgCl_2$), resuspended to $10^9$ CFU/mL, and combined with 100 μM nitrocefin (Sigma-Aldrich). Absorbance at 486 nm was recorded every minute for an hour using a BioTek Eon spectrophotometer.

## Monitoring intracellular pH with BCECF

Overnight cultures were diluted 1:50 and regrown to early log phase ($OD_{600}$ 0.4–0.5). BCECF (2',7'-Bis-(2-Carboxyethyl)-5-(and-6)-Carboxyfluorescein, Acetoxymethyl Ester (BCECF-AM) (Molecular Probes) was added to a final concentration of 10 μM and incubated at 37˚C in a rotator for one hour. Cells were diluted 1:4 and pipetted into a black 96-well plate (Greiner). After five minutes of equilibration, compounds were added and fluorescence (ex490/em535 nm and ex440/em535 nm) was monitored every 2.5 minutes for 20 minutes using a BioTek Synergy H1 plate reader. BCECF fluorescence was calibrated at 7 pHs between 5.5 and 8 (every 0.5 pH): $pH = pK_a - \log(I_{490}/I_{440})$; the $pK_a$ of BCECF is 6.97 [122].

## Resazurin assays

Overnight cultures were subcultured 1:50 in fresh LB and grown to mid-log phase ($OD_{600}$ 0.4–0.6). Cells (200 μL per well) were transferred to a black 96-well plate (Greiner) containing compound. Resazurin (alamarBlue, Invitrogen) was added to a final concentration of 100 μg/mL. The plate was incubated with shaking in the dark at room temperature for five minutes. Fluorescence readings were taken every five minutes for thirty minutes (ex570/em650 nm) using a BioTek Synergy H1 plate reader.

## ATP measurements

Intracellular ATP levels were measured using BacTiter-Glo Microbial Cell Viability Assay (Promega) according to the manufacturer's instructions. Overnight cultures were subcultured in fresh LB and grown to an $OD_{600}$ of 0.35–0.45. Cells (100 μL) were added to 2 μL of compound in a 96-well plate and incubated for 10 minutes at 37°C with agitation. Reagent (100 μL) was added and incubated in the dark with agitation for 5 minutes. Luminescence was read on a BioTek Synergy H1 plate reader.

## Propidium iodide membrane barrier assays

Overnight cultures were subcultured into fresh LB and grown to mid-log phase ($OD_{600}$ 0.4–0.6). Compound, DMSO or 0.008% SDS was added to the desired concentration, and cultures were sampled at 0, 5, 10, 15, and 30 minutes. Five minutes before a sample was harvested, propidium iodide (Life Technologies) was added to a final concentration of 10 μg/mL. Cells were pelleted, washed twice, resuspended in PBS, and monitored (ex535/em617 nm) using a BioTek Synergy H1 plate reader.

## Membrane fluidity assays with Laurdan

Overnight cultures were subcultured 1:50 in fresh LBg (LB with 0.2% glucose) and grown to mid-log phase ($OD_{600}$ 0.4–0.6). Laurdan (Invitrogen) was added to a final concentration of 10 μM and incubated at 37°C with rotation for 30 minutes. Cells were harvested by centrifugation, washed three times, and resuspended in prewarmed PBSg (PBS with 0.2% glucose). Cells (200 μL) were transferred to a black 96-well plate (Greiner) and a monitored (ex360/em450 and 500 nm) on a BioTek Synergy H1 plate reader. Baseline fluorescence was recorded for five minutes prior to addition of compound. Fluorescence was recorded for 25 additional minutes. Laurdan generalized polarization (GP) was calculated: $GP = (I_{460} - I_{500})/(I_{460} + I_{500})$.

## Bacterial SR-SIM fluorescence microscopy

Cultures were grown in LB at 37°C overnight, diluted 1:100 into LB, and grown to an $OD_{600}$ of 0.4–0.5. Nile red was added to a final concentration of 30 μM and incubated at 37°C for 10 minutes. Cells were harvested by centrifugation at 10,000 $x$ $g$ for 30 seconds and the supernatant was carefully removed. Cell pellets were resuspended in 100 μL of FluoroBrite DMEM media and deposited (3 μL) onto an agar pad (20% LB, 2% agarose) containing either 1x MIC JD1 or 1.75% DMSO. Cells were covered with a 1.5H glass coverslip and imaged. Briefly, cells were time-lapsed imaged in 3D-SIM mode using a Nikon structured illumination super-resolution microscope with a 100x/1.49NA Oil SR Apo TIRF WD 0.12 (mm) objective and/or iXon X3 EM-CCD 512 X 512 16-bit camera (ex561nm/em624/40 nm with a standard Texas red filter). Images were acquired at the time intervals shown and reconstructed using Nikon Elements SR-SIM analysis software with the default reconstruction parameters. Both Nikon Perfect Focus and manual focusing were used to find the best focal plane during acquisition.

## LDH assays

The CyQUANT LDH Cytotoxicity Kit (Invitrogen) was used according to the manufacturer's instructions.

## RAW 264.7 membrane integrity microscopy

RAW 264.7 cells between passages one and six were grown in complete DMEM to a confluency of 70–90%. Cells were scraped, washed once, resuspended, and diluted to $5x10^5$ cells/

mL. Cells (100 μL) were added to a black 96-well glass-bottomed plate (Brooks Life Sciences) and incubated for 23.5 hours at 37˚C with 5% $CO_2$. At 23.5 hours the medium was exchanged for 100 μL of 3.4 μM Nile red in FluoroBrite DMEM and incubated for 30 minutes. The medium was exchanged for 150 μL of 0.314 μM Nile red in FluoroBrite DMEM. Compounds were added in 50 μL to obtain the desired concentration. The final volume per well was 200 μL, with 0.235 μM of Nile red and 0.32% DMSO. Cells were time-lapse imaged on a Yokogawa CellVoyager CV1000 Confocal Scanner System with a 40x/0.6NA objective (ex561/em617/73 nm) in an environmentally controlled chamber every hour for 17 hours. Two fields per well were imaged with each field comprised of five images sampled over a z-dimension of 15 μm. The resulting images were converted into maximum intensity projections and used to evaluate morphological changes in the cell membrane over time.

## Mitochondrial membrane determination with TMRM

Experiments were performed with RAW 264.7 cells between passages one and six. Cells were grown in complete DMEM to a confluency of 70–90%. Cells were scraped, washed once, resuspended and diluted to a final concentration of $5x10^5$ cells/mL. Cells (100 μL) were transferred to a 96-well black glass-bottomed plate (Brooks Life Sciences) and incubated for 23.5 hours at 37˚C with 5% $CO_2$. At 23.5 hours, the medium was exchanged for a 100 μL of FluoroBrite DMEM containing 100 nM of TMRM. Thirty minutes later, the medium was exchanged for 150 μL of FluoroBrite DMEM. Cells were time-lapse imaged on a Yokogawa CellVoyager CV1000 Confocal Scanner System with a 20x/0.75NA objective and an environmentally controlled chamber for 30 minutes with images acquired every 10 minutes. Compounds were then added to the plate with a multichannel pipet as 50 μL aliquots to obtain the desired concentration and a final volume of 200 μL. All wells contained a final concentration of 0.4% DMSO except for the CCCP control, which required 0.5% DMSO to remain in solution. Cells were imaged every 10 minutes for 80 minutes. Two fields per well were imaged with each field comprising five images sampled over a z-dimension of 15 μm. Images were converted into maximum intensity projections and the TMRM foreground signal was extracted via a MATLAB R2018a (MathWorks) script and normalized to time zero for each field.

## Murine infections

Female C57Bl/67–8-week-old mice were intraperitoneally (IP) inoculated with $8 x 10^3$ CFU, and the infectious dose was verified by plating for CFU. Mice were IP-treated with 100 μL of vehicle (70% DMSO) or 1 mg/kg of JD1 in 100 μL of vehicle at 10 minutes and 24 hours post-infection based on a previously established protocol [22]. Dosages were extrapolated from LDH toxicity and SAFIRE $IC_{50}$ assays, according to approved IACUC protocols [22]. At 1 mg/kg uninfected mice had no apparent adverse reactions over two days of treatment. At 48 hours post-inoculation, infected animals were euthanized by $CO_2$ asphyxiation, followed by cervical dislocation. Spleen and liver were collected, homogenized in 1 mL PBS and serially diluted for plating to enumerate CFU. Liver *P*-value is 0.25.

## Liposome preparation

Lipids used in this work were purchased from Avanti Polar Lipids. For liposomes made from unlabeled lipid mixtures, 1-palmitoyl-2-oleoyl-sn-glycero-3-phosphocholine (POPC), 1-palmitoyl-2-oleoyl-sn-glycero-3-phosphoethanolamine (POPE), 1-palmitoyl-2-oleoyl-sn-glycero-3-phosphoserine (POPS), and cholesterol were mixed in a molar ratio of 60:20:10:10. The various molar ratios of POPS (0, 20, and 30%) and cholesterol (0%) were balanced by POPC. For liposomes made from *E. coli* lipid extracts, phosphatidylethanolamine (PE),

phosphatidylglycerol (PG), and cardiolipin (CL) were mixed in a ratio (wt/wt%) of 67:23.2:9.8. Liposomes were prepared by detergent dilution and reconstituted in the presence of 50 mM sulforhodamine B (Sigma-Aldrich). Free sulforhodamine B was removed by overnight dialysis using Novagen dialysis tubes against the reconstitution buffer (100 mM KCl, 25 mM HEPES [pH7.4] with 10% glycerol) followed by liposome flotation on a Nycodenz gradient [62].

### Liposome leakage assay

A standard liposome leakage assay contained 40 μL of reconstitution buffer without glycerol, 5 μL of sulforhodamine B-loaded liposomes, and 5 μL of the desired concentration of JD1. The leakage assay was conducted in a 96-well microplate at 37°C, and sulforhodamine B fluorescence (ex565/em585 nm) was measured every 2 minutes in a BioTek Synergy H1 plate reader. Ten μL of 10% CHAPSO (Sigma-Aldrich) was added to each sample at the end of the reaction [123]. Leakage data were presented as the percentage of maximum fluorescence change based on three biological replicates.

## Supporting information

**S1 Fig. JD1 anti-bacterial activity is not synergistic in broth or in macrophages with gentamicin.** A) *S.* Typhimurium in LB with 0.5 μg/mL PMB was grown overnight in the presence of JD1 and gentamicin. The darkest blue represents an $OD_{600}$ of 0.94 and white represents 0. The fractional inhibitory concentration index (FICI) was determined using the following equation: 7.5/15 + 0.313/0.625 = 0.5 +0.5 = 1, indicating no synergy. A representative of three biological replicates is shown. B-D) SAFIRE (B) and CFU (C, D) assays in which our standard concentration of gentamicin (40 μg/mL) was retained or switched for a lower concentration (10 μg/mL) at 2 hours after infection, just prior to compound addition. The SAFIRE assay used strain SL1344 *rpsM*::*gfp*, whereas the CFU assays used SL1344 or 14028 as indicated.
(TIF)

**S2 Fig. The growth inhibitory activity of JD1 is not diminished by the presence of an iron chelator, and JD1 is potent against log- and lag-phase cells, but not early stationary phase cells.** A) Treatment with the iron chelator deferasirox (DFS) does not rescue growth inhibition by JD1 (1x MIC) in LB with 0.5 μg/mL PMB. Mean and SEM of four biological replicates performed with technical triplicates. B-E) JD1 inhibits growth and kills cells in log-phase and lag-phase but not in early stationary phase. Overnight cultures were diluted 1:100 into LB with 0.5 μg/mL PMB and grown to the following ODs prior to the addition of DMSO or JD1 at 2x MIC (28 μM): ~0.1 (lag phase; B, E), 0.4–0.5 (log phase; C, F), or 1.0–1.4 (early stationary phase; D, G). (B-D) Cultures were monitored for $OD_{600}$. (E-G) Cultures were plated for enumeration of CFU. Mean and SEM of three biological replicates.
(TIF)

**S1 Video. DMSO control from live imaging in Fig 1D.** RAW 264.7 cells were infected with SL1344 *sifB*::*gfp*. Forty-five minutes after infection the medium was exchanged for FluoroBrite DMEM with 40 μg/mL of gentamicin and incubated for an additional 45 minutes. Compounds were added to the indicated concentration and imaged every 30 minutes for 17 hours in an environmentally controlled chamber. Videos are representative of six fields of view acquired per well for each of three biological replicates performed in technical triplicate. Scale bar is 20 μm.
(MOV)

**S2 Video. 0.6 μM JD1 treatment from live imaging in Fig 1D. Cells were processed as described for S1 Video.**
(MOV)

**S3 Video. 0.18 μM JD1 treatment from live imaging in Fig 1D. Cells were processed as described for S1 Video.**
(MOV)

**S4 Video. 0.56 μM JD1 treatment from live imaging in Fig 1D. Cells were processed as described for S1 Video.**
(MOV)

**S5 Video. 1.7 μM JD1 treatment from live imaging in Fig 1D. Cells were processed as described for S1 Video.**
(MOV)

**S6 Video. 5.0 μM JD1 treatment from live imaging in Fig 1D. Cells were processed as described for S1 Video.**
(MOV)

**S7 Video. 0.25 μM JD1 treatment from live imaging in Fig 7A.** RAW 264.7 cells were stained with 3.4 μM Nile red for 30 minutes, washed, and then supplemented with FluoroBrite DMEM containing 0.235 μM Nile red. Cells were treated with DMSO or the indicated concentrations of JD1 and imaged every hour for 17 hours in an environmentally controlled chamber. Videos are representative of two fields of view acquired per well for each of three biological replicates performed in technical triplicate. Scale bar is 20 μm.
(AVI)

**S8 Video. 0.5 μM JD1 treatment from live imaging in Fig 7A. Cells were processed as described for S7 Video.**
(AVI)

**S9 Video. 1.0 μM JD1 treatment from live imaging in Fig 7A. Cells were processed as described for S7 Video.**
(AVI)

**S10 Video. 2.0 μM JD1 treatment from live imaging in Fig 7A. Cells were processed as described for S7 Video.**
(AVI)

**S11 Video. 4.0 μM JD1 treatment from live imaging in Fig 7A. Cells were processed as described for S7 Video.**
(AVI)

**S12 Video. 8.0 μM JD1 treatment from live imaging in Fig 7A. Cells were processed as described for S7 Video.**
(AVI)

**S13 Video. 16 μM JD1 treatment from live imaging in Fig 7A. Cells were processed as described for S7 Video.**
(AVI)

**S14 Video. 32 μM JD1 treatment from live imaging in Fig 7A. Cells were processed as described for S7 Video.**
(AVI)

**S15 Video. 64 μM JD1 treatment from live imaging in Fig 7A. Cells were processed as described for S7 Video.**
(AVI)

## Acknowledgments

We thank all the members of the Detweiler laboratory for insightful discussions and technical help. We are grateful to P. Muhlrad and J. Villanueva for comments on the manuscript. We thank the MCDB Light Microscopy Facility at the University of Colorado Boulder and in particular James Orth for expert guidance. We also thank the University of Colorado BioFrontiers Institute Next-Gen Sequencing Core Facility for Illumina sequencing and library construction, and K. Hiller for expertise in analyzing sequencing data. Publication of this article was funded by the University of Colorado Boulder Libraries Open Access Fund.

## Author Contributions

**Conceptualization:** Jamie L. Dombach, Corrella S. Detweiler.

**Data curation:** Jamie L. Dombach, Joaquin L. J. Quintana, Toni A. Nagy, Chun Wan, Amy L. Crooks, Haijia Yu.

**Formal analysis:** Jamie L. Dombach, Joaquin L. J. Quintana, Toni A. Nagy, Amy L. Crooks.

**Funding acquisition:** Edward W. Yu, Jingshi Shen, Corrella S. Detweiler.

**Investigation:** Jamie L. Dombach, Joaquin L. J. Quintana, Toni A. Nagy, Chun Wan, Amy L. Crooks, Haijia Yu, Chih-Chia Su.

**Methodology:** Jamie L. Dombach, Joaquin L. J. Quintana, Toni A. Nagy, Chun Wan, Amy L. Crooks, Haijia Yu, Edward W. Yu, Jingshi Shen, Corrella S. Detweiler.

**Project administration:** Jamie L. Dombach, Edward W. Yu, Jingshi Shen, Corrella S. Detweiler.

**Resources:** Edward W. Yu, Jingshi Shen, Corrella S. Detweiler.

**Software:** Joaquin L. J. Quintana.

**Supervision:** Jamie L. Dombach, Edward W. Yu, Jingshi Shen, Corrella S. Detweiler.

**Validation:** Jamie L. Dombach, Edward W. Yu, Jingshi Shen, Corrella S. Detweiler.

**Visualization:** Jamie L. Dombach, Joaquin L. J. Quintana, Toni A. Nagy, Chun Wan, Amy L. Crooks, Chih-Chia Su.

**Writing – original draft:** Jamie L. Dombach, Corrella S. Detweiler.

**Writing – review & editing:** Jamie L. Dombach, Joaquin L. J. Quintana, Toni A. Nagy, Chun Wan, Haijia Yu, Edward W. Yu, Jingshi Shen, Corrella S. Detweiler.

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
