## [Decision Letter · Decision Letter 0]

16 Jul 2020

Dear Dr. Detweiler,

Thank you very much for submitting your manuscript "A small molecule that mitigates bacterial infection disrupts Gram-negative cell membranes and is inhibited by cholesterol and neutral lipids" for consideration at PLOS Pathogens. As with all papers reviewed by the journal, your manuscript was reviewed by members of the editorial board and by several independent reviewers. In light of the reviews (below this email), we would like to invite the resubmission of a significantly-revised version that takes into account the reviewers' comments.

We cannot make any decision about publication until we have seen the revised manuscript and your response to the reviewers' comments. Your revised manuscript is also likely to be sent to reviewers for further evaluation.

Sincerely,

Leigh Knodler

Guest Editor

PLOS Pathogens

Renée Tsolis

Section Editor

PLOS Pathogens

Kasturi Haldar

Editor-in-Chief

PLOS Pathogens

orcid.org/0000-0001-5065-158X

Michael Malim

Editor-in-Chief

PLOS Pathogens

orcid.org/0000-0002-7699-2064

Reviewer's Responses to Questions

**Part I - Summary**

Reviewer #1: Dombach et al report the characterization of a small molecule they call JD1. They present compelling evidence that the drug somehow disrupts that inner membrane of Salmonella and E. coli; eukaryotic membranes are protected by their different overall charge and the presence of cholesterol. The authors use a wide variety of techniques to support their conclusions and the experiments are well done. My comments focus on clarity.

Reviewer #2: The manuscript “A small molecule that mitigates bacterial infection disrupts Gram-negative cell membranes and is inhibited by cholesterol and neutral lipids” is well-written and describes the mechanism of action of a compound, JD1, that was previously identified in a chemical screen for inhibitors of Salmonella within macrophages. Through in vitro experiments using a combination of different media and characterized bacterial mutants, the authors determine that the compound alters membrane morphology of two Gram-negative organisms. Furthermore, the compound is shown to be a substrate for efflux by the major drug efflux pump AcrAB. The data demonstrate the effects of the compound on membranes are more pronounced in bacteria because of the more highly negative charge of bacterial membranes as compared with eukaryotic membranes. The compound’s mechanism of action is novel for inhibitors of Gram-negative organisms.

The major limitation to the relevance of the conclusions is a lack of correlation of the in vitro mechanistic data regarding bacterial membranes and efflux with intracellular drug efficacy. As a minor point, the in vivo drug effects in a mouse model of salmonellosis are modest.

Reviewer #3: In Dombach et al., the authors explore the mechanism of action of the small molecule JD1 against Salmonella Typhimurium (STm). JD1 was identified using a screen which looks for inhibition of STm growth during infection of RAW264.7 macrophage cells, which has been previously described by this group. Here they show that the IC95 of JD1 is 1.2 µM in RAW cells as compared to an MIC of 89 µM in a media that resembles the intracellular (vacuolar) environment and 77 µM in a nutrient poor M9 media. They then use a number of in vitro assays to show that environments that weaken or damage the bacterial cell membrane make STm more susceptible to JD1. In LB, polymyxin B reduced the MIC to 14 µM and this condition was then used in further study including showing the ability of JD1 to affect bacterial membrane fluidity. However, at a very similar concentration (16 µM), JD1 can affect eukaryotic cell membranes and leads to cell death at 32 µM. In a murine model JD1 reduced tissue colonization by STm without “obvious whole-animal toxicity”, although mice were treated only twice with the drug and no toxicity studies per se were carried out. A final section of the study addresses the possibility that a combination of cholesterol and neutral lipids may protect mammalian cells from JD1. Overall the experiments are well done, although additional experiments with different cells or cell lines and at least one other STm strain are needed to show that the antibacterial activity of JD1 is not an artifact of system. Furthermore, the antibacterial activity of JD1 is rather weak and, as the authors themselves stated, it is not a strong candidate for further development.

**Part II – Major Issues: Key Experiments Required for Acceptance**

Reviewer #1: No additional experiments required

Reviewer #2: The MIC95 of JD1 for the Salmonella ∆acrAB mutant and the WT in different media compositions is approximately 100x that found within macrophages (as stated in Table 1). This finding suggests that there may be a significant dichotomy in the mechanism of JD1 in macrophages versus that found in media and may diminish the conclusions regarding membrane fluidity and efflux. A demonstration of the effect of efflux and membrane composition, either through chemical or genetic means, on the efficacy of JD1 against intracellular Salmonella would demonstrate relevance of mechanistic studies on inhibition of intracellular Salmonella. Furthermore, these discrepancies merit comment in the discussion.

Correlation of the in vitro derived mutants with improved compound resistance in macrophages would strengthen the conclusions of the manuscript regarding efflux.

Together the data demonstrate an effect of JD1 on membranes of Gram-negative bacteria, mitochondria, and eukaryotic cells. While the concentrations needed to cause gross changes to macrophages are higher than those needed to reduce intracellular Salmonella, it is unclear whether minor membrane damage caused by JD1 could increase permeability of macrophages to gentamicin in the assay buffer and therefore the efficacy of JD1 could represent sensitivity of intracellular Salmonella to gentamicin rather than to JD1. Experiments to rule out this potential confounding factor would strengthen the conclusions of the manuscript regarding the efficacy of JD1 against intracellular Salmonella.

Reviewer #3: 1) Although the intracellular activity of JD1 is of critical importance to the study, this was only demonstrated in RAW264.7 cells, a macrophage-like cell line that is quite permissive for intracellular replication of STm. What about in primary mouse macrophages? Or other relevant host cells such as epithelial cells. The role of antimicrobial peptides could be more thoroughly investigated using macrophages from mice deficient in the production of e.g. CRAMP or by the addition of purified peptides to the culture.

2) The strain of STm used here, SL1344, is a histidine auxotroph that has impaired growth in macrophages (Henry et al Molecular Microbiology (2005) 56(1), 252–267). Does this contribute to the antimicrobial activity of JD1? Are other STm strains e.g. 14028 also sensitive to JD1? What happens if supplemental histidine, which relieves the growth defect of intracellular SL1344, is added to the culture media?

3) Is the effect of JD1 growth phase dependent? When testing the activity of JD1 against intracellular bacteria the drug was added at 2 h pi a time at which the bacteria are not actively replicating. The inoculum was prepared from overnight (stationary?) cultures and the bacteria do not replicate intracellularly until >4 h pi. In contrast, in the in vitro assays JD1 was primarily added during logarithmic growth. What is the effect of adding JD1 to cultured bacteria during lag phase rather than logarithmic?

**Part III – Minor Issues: Editorial and Data Presentation Modifications**

Reviewer #1: 1. Fig 1. You need to more clearly explain how you calculate GFP+ Macrophage Area. You describe how you acquire the data, but not how you get the underlying values which are then converted to a percent. Likewise for the Integrated Density.

2. Table 1. Why is the IC95 given in two different units? Why is PMB listed ad “ug” and PMBN listed as “ug/ml”?

3. It is not clear that the isothermal calorimetry with purified AcrB adds much to the paper. Given that the protein is transmembrane, you must at least describe its purification and your confidence in it being in a physiological conformation. What is the binding affinity for some known substrate?

4. Line 226 and 634. The methods sentence is awkwardly worded. It can be interpreted as saying that you took one single colony and inoculated 6 cultures. I assume that each independent culture was inoculated with a separate colony. Reword.

5. Line 246. Did you ever test any of the efflux mutants in the macrophage assay? Not a big deal, just an interesting question.

6. Line 254. “With a LogP value of 12.5…” You need a reference here or some explanation of what LogP means. You explain it later, but not here.

Reviewer #2: The MIC95 data presented in Table 1 are contradicted in the text and figures. Line 143 states MIC95 of 0.63 µM within macrophages whereas table 1 indicates a MIC95 of intracellular Salmonella of 1.2 µM by GFP fluorescence. Imaging of membrane morphology states usage of 1X MIC (line 312) and figure 6 shows 14 µM JD1 whereas the MIC95 listed in table 1 are 25.9 and 20.06 µM for each E. coli strain shown. Please clarify in methods or legends for figures 2, 4, and 5 the concentrations of JD1 used for each data set.

Lines 239-40: “Upon removal of CCCP and the addition of glucose 5 of 6 mutant strains exported…” contradicts data presented in Table 2 that demonstrates increased efflux for 2 of 6 mutants upon addition of glucose. Please correct and comment upon potential mechanism for lack of response to glucose on efflux.

Lines 243-4: Are the resistant mutants 1-6 identified through in vitro acclimation resistant to the effects of JD1 in macrophages?

Line 353: Are clinical data available for mouse infections? Was there evidence of tissue damage in treated animals?

Lines 643-9: The sequencing results for mutants 1 and 4 should be described in the results.

Line 696: Please describe calculations used to determine intracellular pH.

Line 805-6: Thank you for indicating approval of compound administration dosages in animal use protocols. Please describe the rationale for JD1 dose and dosage regimen for mice. In what volume were drug and compound administered?

Line 809 indicates collection of clinical data from mice. Please display these data in the manuscript.

Line 1141: Please indicate the media in which the ∆acrAB mutant was evaluated.

Figure 4G: Y-axis describes data normalized to DMSO, but data suggest that DMSO increases above 1 with time. Please clarify.

Reviewer #3: 1) The manuscript would benefit from editing for brevity and clarity. The following are provided as examples:

- In the introduction, the paragraph about the SAFIRE screen is extraneous as it has already been published and could be reduced to a sentence citing the original paper.

- The first two paragraphs of the intro are somewhat redundant.

- The lines 61-63 from the summary state, “This is the first compound, to our knowledge, that preferentially targets the cell membranes of Gram-negative bacteria and reduces bacterial infection of animals.” But lines 491-492 in the conclusion state “At least one lipophilic compound, metergoline, has a demonstrated ability to damage the cell membrane of Gram-negative bacteria and to inhibit infection.” These seem contradictory

- Within the results section there are some inconsistencies that make it hard to follow or interpret the data. In particular, it is not always clear which membrane or membranes are being analyzed for each assay and ultimately which membrane JD1 is affecting. Also, the use of MIC is not consistent. Defined as the concentration at which 95% of growth was inhibited, lines 162-13, in other places they used phrases such as “completely unable to grow” (line 173) or “conc. Needed to inhibit growth” (line 186).

- Several sections in the Methods are not fully clear. For example, in the evolution of resistance, how long were bacterial cultures grown? How was it determined if there was “visible growth”? OD600? Also, the text states that growth at 3X MIC took approximately 12 generations (line 637). Should this be passages rather than generations?

2) The SifB::GFP reporter used in the SAFIRE screen has also been used here to demonstrate that JD1 blocks replication of intracellular bacteria. However, another inhibition of SPI2 induction within intracellular bacteria would also block GFP expression. Why not use a constitutive GFP reporter that would show the total intracellular population?

3) Several pieces of data seem to be missing from the manuscript.

- The iron chelator deferasirox

- Hoechst staining

- Pictures or quantification of liver abscess

4) There should be consistency and clarity in their statistics throughout the paper. Standard deviation should be used to convey the spread of variability of the data, not SEM, which tends to just show smaller error bars. Three biological replicates should be done for every figure. Some of the figures are lacking the statistical test used in the figure legend. Finally, Students t-test should not be used when more than two groups are being compare e.g. Fig 8

PLOS authors have the option to publish the peer review history of their article (what does this mean?). If published, this will include your full peer review and any attached files.

Reviewer #1: No

Reviewer #2: No

Reviewer #3: No
---

## [Decision Letter · Decision Letter 1]

1 Nov 2020

Dear Corrie,

We are pleased to inform you that your manuscript 'A small molecule that mitigates bacterial infection disrupts Gram-negative cell membranes and is inhibited by cholesterol and neutral lipids' has been provisionally accepted for publication in PLOS Pathogens.

Best regards,

Leigh Knodler

Guest Editor

PLOS Pathogens

Renée Tsolis

Section Editor

PLOS Pathogens

Kasturi Haldar

Editor-in-Chief

PLOS Pathogens

orcid.org/0000-0001-5065-158X

Michael Malim

Editor-in-Chief

PLOS Pathogens

orcid.org/0000-0002-7699-2064

Dear Corrie,

Thank you for your attention to the reviewer's suggestions to improve the quality of your original submission. All reviewer's are satisfied with the revised manuscript, as am I. Could you please be sure to correct the minor issues in the text indicated by Reviewer 2 in the editing stage.

Best wishes, Leigh

Reviewer Comments (if any, and for reference):

Reviewer's Responses to Questions

**Part I - Summary**

Reviewer #2: The manuscript reports a small molecule, JD1, that inhibits Salmonella replication in two different eukaryotic cell types and reduces systemic colonization in mice. Resistance to JD1 is linked to the major drug efflux pump AcrAB-TolC and to the integrity of the outer membrane. The data presented suggest a role for JD1 in increasing fluidity of the inner membrane and allowing for loss of integrity and reducing intracellular ATP. JD1 also displays toxicity to eukaryotic cells, although at higher concentrations than needed for bacterial killing, as a result of membrane composition.

The revised manuscript is improved for clarity and the conclusions are justified by the data presented. The added experiments have facilitated acceptance of the authors' conclusions regarding compound function. Furthermore, the revised discussion is improved for clarity and now provides a comprehensive discussion of the potential mechanism of action of the compound within eukaryotic cells vs in media alone. The comparison of JD1 with other cell membrane altering antibiotics places it into context as a potential lead in drug discovery.

Reviewer #3: The authors have made a good attempt to answer all the reviewers comments and the paper is much improved. I particularly appreciate that they have added another Salmonella strain and a different host cell type. Unfortunately animal experiments, or experiments with mouse macrophages, could not be done due to the limitations imposed in response to the pandemic. Although these would have added significantly to the study it is still a well done and interesting story.

**Part II – Major Issues: Key Experiments Required for Acceptance**

Reviewer #2: None

Reviewer #3: (No Response)

**Part III – Minor Issues: Editorial and Data Presentation Modifications**

Reviewer #2: Line 211 refers to figure S1A although data are in figure S2A.

Line 240 “lacking acrAB or tolC, respectively, were sensitive to JD1…” should read “more sensitive to JD1…”

Line 338 states that JD1 at 2x MIC increases PI fluorescence within 5 minutes, although the asterisks in figure 5A indicate that PI fluorescence increases within 10 minutes. Please align the figure with the text.

Line 1312 indicates gramcidin treatment prior to measurement of intracellular ATP measurement although data are not shown in figure 5H. Please correct legend.

Reviewer #3: (No Response)

PLOS authors have the option to publish the peer review history of their article (what does this mean?). If published, this will include your full peer review and any attached files.

Reviewer #2: No

Reviewer #3: No

---

## [Editor Report · Acceptance letter]

23 Nov 2020

Dear Dr. Detweiler,

We are delighted to inform you that your manuscript, "A small molecule that mitigates bacterial infection disrupts Gram-negative cell membranes and is inhibited by cholesterol and neutral lipids," has been formally accepted for publication in PLOS Pathogens.

Best regards,

Kasturi Haldar

Editor-in-Chief

PLOS Pathogens

orcid.org/0000-0001-5065-158X

Michael Malim

Editor-in-Chief

PLOS Pathogens

orcid.org/0000-0002-7699-2064